# Trends in self-citation rates in high-impact neurology, neuroscience, and psychiatry journals

Matthew Rosenblatt[1]*, Saloni Mehta[2], Hannah Peterson[2], Javid Dadashkarimi[3], Raimundo Rodriguez[4], Maya L Foster[1], Brendan D Adkinson[4], Qinghao Liang[1], Violet M Kimble[4], Jean Ye[4], Marie C McCusker[4], Michael C Farruggia[4], Max J Rolison[5], Margaret L Westwater[2], Rongtao Jiang[2], Stephanie Noble[2,6,7], Dustin Scheinost[1,2,4,5,8]

[1]Department of Biomedical Engineering, Yale University, New Haven, United States; [2]Department of Radiology & Biomedical Imaging, Yale School of Medicine, New Haven, United States; [3]Department of Computer Science, Yale University, New Haven, United States; [4]Interdepartmental Neuroscience Program, Yale University, New Haven, United States; [5]Child Study Center, Yale School of Medicine, New Haven, United States; [6]Department of Bioengineering, Northeastern University, Boston, United States; [7]Department of Psychology, Northeastern University, Boston, United States; [8]8Department of Statistics & Data Science, Yale University, New Haven, United States

*For correspondence: matthew.rosenblatt@yale.edu

## eLife Assessment

This study examines how self-citations in selected neurology, neuroscience, and psychiatry journals differ according to seniority, geography, gender and subfield. The evidence supporting the claims is **convincing**, and the article is a **valuable** addition to the literature on self-citations.

**Abstract** Citation metrics influence academic reputation and career trajectories. Recent works have highlighted flaws in citation practices in the Neurosciences, such as the under-citation of women. However, self-citation rates—or how much authors cite themselves—have not yet been comprehensively investigated in the Neurosciences. This work characterizes self-citation rates in basic, translational, and clinical Neuroscience literature by collating 100,347 articles from 63 journals between the years 2000–2020. In analyzing over five million citations, we demonstrate four key findings: (1) increasing self-citation rates of Last Authors relative to First Authors, (2) lower self-citation rates in low- and middle-income countries, (3) gender differences in self-citation stemming from differences in the number of previously published papers, and (4) variations in self-citation rates by field. Our characterization of self-citation provides insight into citation practices that shape the perceived influence of authors in the Neurosciences, which in turn may impact what type of scientific research is done and who gets the opportunity to do it.

## Introduction

Citations are often used as a proxy for how well a researcher disseminates their work, which is important both for spreading knowledge and establishing a scientific reputation (*Petersen et al., 2014*). Furthermore, citation counts and other metrics like the h-index are critical for hiring and

promotion in an increasingly tenuous academic job market (*Abbott, 2010*; *Else, 2021*; *Holden et al., 2005*), necessitating a thorough examination of citation practices across research fields. Existing investigations of citation practices have found, for instance, false inflation of impact factors by specific journals (*Van Noorden, 2013*). Others have demonstrated under-citation of racial and ethnic minority groups *Bertolero et al., 2020* and women (*Dworkin et al., 2020*; *Chatterjee and Werner, 2021*; *Fulvio et al., 2021*), including three studies specific to the Neuroscience literature (*Bertolero et al., 2020*; *Dworkin et al., 2020*; *Fulvio et al., 2021*). These examples of citation manipulations and biases underscore the importance of comprehensively investigating citation practices in the broader Neuroscience literature.

Self-citation, or how frequently authors cite themselves, remains an understudied citation practice in the Neuroscience literature. Self-citation can be calculated from two different perspectives: (1) as the proportion of an author's total citations that come from their own works (*Ioannidis et al., 2019*; *Aksnes, 2003*) or (2) as the proportion of an author's references on which they are also an author (*Snyder and Bonzi, 1998*). Since the former accounts for the total number of times an author cites themselves (across all papers) divided by the total number of citations the author has received, it helps identify when a particular author only accumulates citations from themselves (*Ioannidis et al., 2019*). However, in this manuscript, we defined self-citation as the latter because one cannot control how much others cite their works. As such, the second definition of self-citation rate may more closely reflect intention in self-citing and will allow for more self-reflection about self-citation practices.

Self-citations may often be appropriate. For example, in a direct follow-up publication, a researcher will need to cite their previous work. Yet, h-indices can be strategically manipulated via self-citation (*Bartneck and Kokkelmans, 2011*), and some scientists may engage in extreme or unnecessary self-citation (*Ioannidis et al., 2019*). While certain citation metrics can be adjusted to remove self-citations, the effect of a single self-citation extends beyond adding one additional citation to an author's citation count. In a longitudinal study of self-citation, *Fowler and Aksnes, 2007* found that each self-citation leads to approximately three additional citations after five years. Given the potential effects of self-citations on various citation metrics that influence career trajectories, a detailed analysis of self-citation rates and trends in the Neuroscience literature could benefit the field.

This work summarizes self-citation rates in Neurology, Neuroscience, and Psychiatry literature across the last 21 years, 63 journals, 100,347 articles, and 5,061,417 citations. We then build upon these calculations by exploring trends in self-citation over time, by seniority, by country, by gender, and by different subfields of research. We further develop models of the number of self-citations and self-citation rate. Finally, we discuss the implications of our findings in the Neuroscience publishing landscape and share a tool for authors to calculate their self-citation rates: https://github.com/mattrosenblatt7/self_citation. (copy archived at *Rosenblatt, 2025*).

## Results

### Data

We downloaded citation information from 157,287 papers published between 2000 and 2020 from Scopus. Articles spanned 63 different journals representing the top Neurology, Neuroscience, and Psychiatry journals (*Appendix 1—table 1*) based on impact factor. After applying our exclusion criteria (see Methods), 100,347 articles and 5,061,417 citations remained.

### Metrics

Using the Scopus database and Pybliometrics API (*Rose and Kitchin, 2019*), we calculated three metrics for each individual paper: First Author self-citation rate, Last Author self-citation rate, and Any Author self-citation rate, where self-citation rate is defined as the proportion of cited papers on which the citing author is also an author. As an example, consider a hypothetical paper by Author A, Author B, and Author C that cites 100 references.

- If Author A is an author on 5 of those references, then the First Author self-citation rate is 5/100=5%.
- If Author C is an author on 10 of those references, then the Last Author self-citation rate is 10/100=10%.

**Table 1.** Self-citation rates in 2016–2020 for First, Last, and Any Authors by field.

| Field | First Author | Last Author | Any Author |
|---|---|---|---|
| Overall | 3.98 (3.87, 4.07) | 8.15 (7.98, 8.30) | 14.41 (13.99, 14.74) |
| Neurology | 4.54 (4.36, 4.70) | 8.87 (8.52, 9.14) | 16.59 (15.85, 17.16) |
| Neuroscience | 3.41 (3.30, 3.51) | 7.54 (7.36, 7.73) | 12.61 (12.29, 12.91) |
| Psychiatry | 4.29 (4.11, 4.43) | 8.41 (8.16, 8.60) | 15.07 (14.48, 15.47) |

- If at least one of Author A, Author B, OR Author C is an author on 18 of the references, then the Any Author self-citation rate is 18/100=18%.

We will use the above definitions of self-citation throughout the remainder of the paper. Furthermore, our estimations via Python code of the above three metrics showed strong agreement with 906 manually scored articles from a subset of Psychiatry journals ($r$=0.98 for First Authors, 0.95 for Last Authors, 0.96 for Any Authors).

We performed 1000 iterations of bootstrap resampling to obtain confidence intervals for all analyses. We additionally performed 10,000 iterations of permutation testing to obtain two-sided p values for all significance tests. All P values are reported after applying the Benjamini/Hochberg (*Benjamini and Hochberg, 1995*) false discovery rate (FDR) correction, unless otherwise specified. Importantly, we accounted for the nested structure of the data in bootstrapping and permutation tests by forming co-authorship exchangeability blocks.

Throughout this work, we characterized self-citation rates with descriptive, not causal, analyses. Our analyses included several theoretical estimands that are descriptive (*Lundberg et al., 2021*), such as the mean self-citation rates among published articles as a function of field, year, seniority, country, and gender. We adopted two forms of empirical estimands. First, we showed subgroup means in self-citation rates. We then developed smooth curves with generalized additive models (GAMs) to describe trends in self-citation rates across several variables.

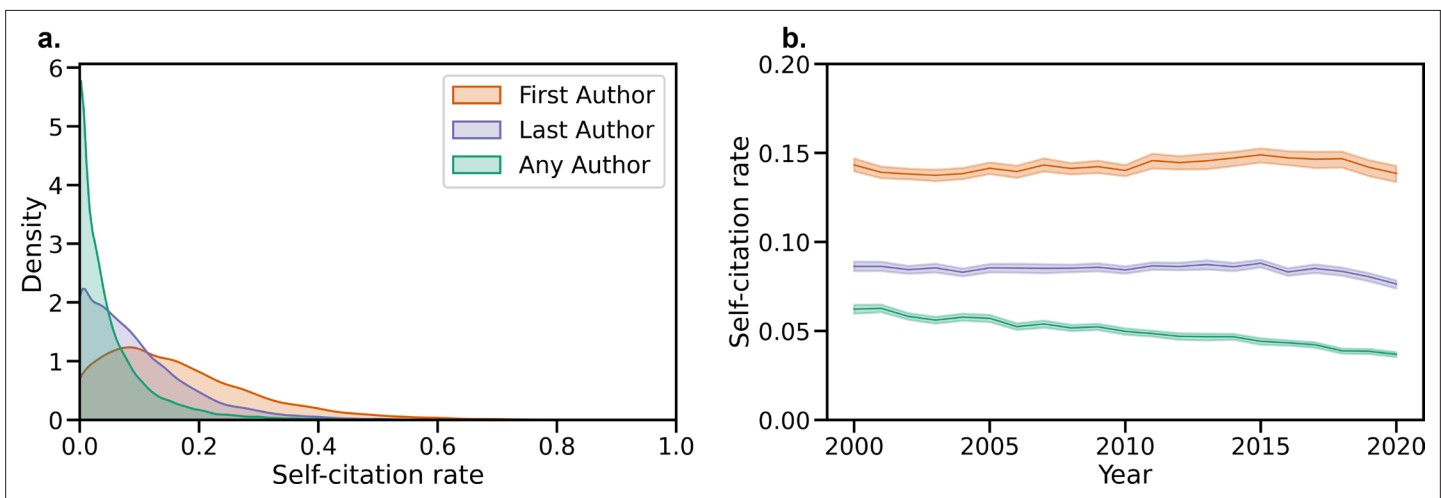

**Figure 1.** Visualizing recent self-citation rates and temporal trends. (**a**) Kernel density estimate of the distribution of First Author, Last Author, and Any Author self-citation rates in the last five years. (**b**) Average self-citation rates over every year since 2000, with 95% confidence intervals calculated by bootstrap resampling.

The online version of this article includes the following figure supplement(s) for figure 1:

**Figure supplement 1.** Temporal trends in First Author, Last Author, and Any Author self-citation rates from 2000 to 2020 in Neurology, Neuroscience, and Psychiatry papers.

## Self-citation rates in 2016-2020

In the last 5 years of our dataset (2016–2020), the overall self-citation rates were 3.98% (95% CI: 3.87%, 4.07%) for First Authors, 8.15% (95% CI: 7.98%, 8.30%) for Last Authors, and 14.41% (95% CI: 13.99%, 14.74%) for Any Authors (*Table 1*). In all fields, the Last Author self-citation rates were significantly higher than that of First Author self-citation rates (p=2.9e-4). Neuroscience had a significantly lower self-citation rate than Neurology and Psychiatry for First, Last, and Any Authors (p's=2.9e-4). We found no significant difference between Neurology and Psychiatry for First Author (p=0.144) and Last Author (p=0.123) self-citation rates. Any Author self-citation rates were significantly higher in Neurology than Psychiatry before correction but nonsignificant after correction (p=0.010). When determining fields by each author's publication history instead of the journal of each article, we observed similar rates of self-citation (*Appendix 2—table 1*). The 95% confidence intervals for each field definition overlapped in most cases, except for Last Author self-citation rates in Neuroscience (7.54% defined by journal vs. 8.32% defined by author) and Psychiatry (8.41% defined by journal vs. 7.92% defined by author).

Although there is no clear rule for what levels of self-citation are 'acceptable,' a histogram of self-citation rates (*Figure 1a*) and a table of self-citation percentiles (*Appendix 2—table 2*) both provide insight into the self-citation levels that are typical in the Neuroscience literature.

## Temporal trends in self-citation rates

Furthermore, self-citation rates have changed since 2000 (*Figure 1*). For example, First Author self-citation rates were 6.22% (95% CI: 5.97%, 6.47%) in 2000 and 3.68% (95% CI: 3.53%, 3.81%) in 2020. First Author self-citation rates decreased at a rate of –1.21% per decade (95% CI: –1.30%, –1.12%), Last Author self-citation rates decreased at a rate of –0.18% per decade (95% CI: –0.31%, –0.05%), and Any Author self-citation rates increased at a rate of 0.32% per decade (95% CI: 0.05%, 0.55%). Corrected and uncorrected p values for the slopes are available in *Appendix 2—table 5*. Further details about yearly trends in self-citation rate by field are presented in *Figure 1—figure supplement 1* and *Appendix 2—table 3*.

## Author seniority and self-citation rate

We also considered that the self-citation rate might be related to seniority. To test this, we calculated each author's 'academic age' as the years between the publication of their first paper (in any author

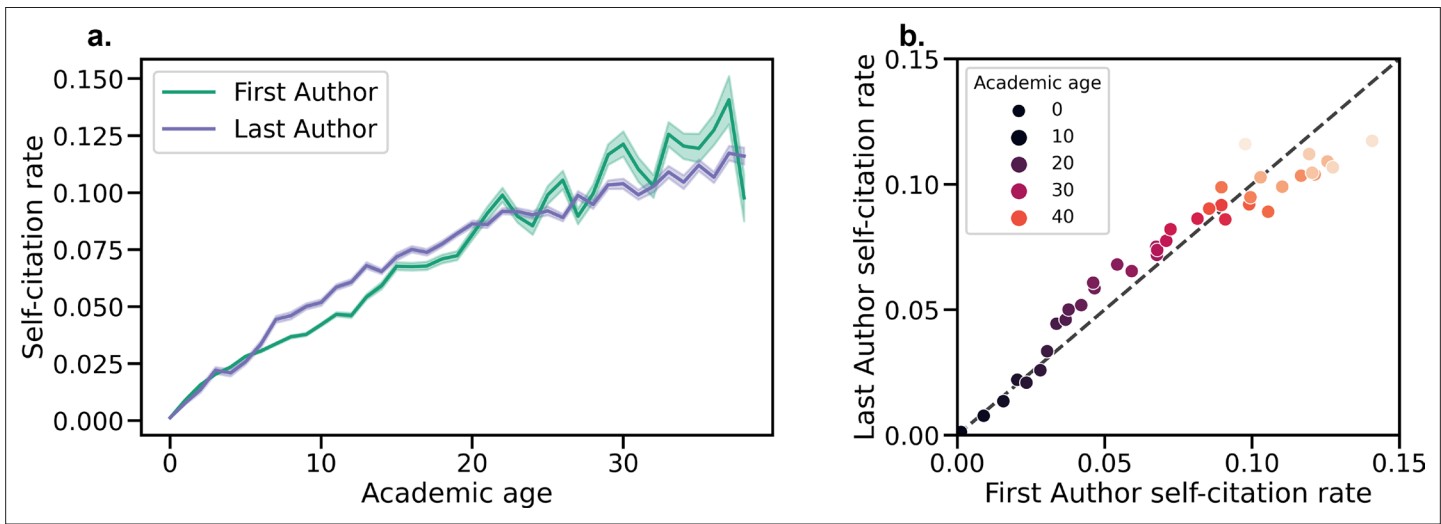

**Figure 2.** Average self-citation rates for each academic age in years 2016–2020. (**a**) Self-citation rate vs. academic age for both First and Last Authors. Shaded regions show 95% confidence intervals obtained via bootstrap resampling. (**b**) Comparison of self-citation rates by academic age for First and Last Authors. For a given academic age, a single point is plotted as (x=First Author self-citation rate for authors of academic age *a*, y=Last Author self-citation rate for authors of academic age *a*). The dashed line represents the y=x line, and the coloring of the points from dark to light represents increasing academic age.

The online version of this article includes the following figure supplement(s) for figure 2:

**Figure supplement 1.** Average of normalized self-citation counts for each academic age in years 2016–2020.

position) and the current paper. For example, if the Last Author of a 2017 paper published their first paper in 1995, their academic age would be 22. We averaged the self-citation rates across each academic age, only including those ages with at least 50 papers in the dataset, and found marked increases in self-citation rate with greater academic age (*Figure 2a*). For instance, at 10 years, the self-citation rate for First Authors is about 5%, while this number increases to over 10% at 30 years. Academic age appears to be a more robust indicator of self-citation than authorship position; for a given academic age, First Author and Last Author self-citation rates are comparable (*Figure 2b*). Analyzing self-citations as a fraction of publication history exhibited a similar trend (*Figure 2—figure supplement 1*). First Authors were even more likely than Last Authors to self-cite when normalized by prior publication history.

## Geographic location and self-citation rate

In addition, we used the country of the affiliated institution of each author to determine the self-citation rate by institution country over the last 5 years (2016–2020). We averaged First Author and Last Author self-citation rates by country and only included countries with at least 50 papers. This analysis is distinct from country self-citation rate because we calculated self-citation at the level of the author, then averaged across countries. In contrast, previous studies have operationalized country self-citation rates as when authors from one country cite other authors from the same country (*Bardeesi et al., 2021*). The results are shown on a map of the world using GeoPandas *Jordahl, 2020*; *Figure 3* and also presented in *Appendix 2—table 4*. Self-citation rates in the highest self-citing countries double that of the lowest for the First and Last Authors. For instance, the First Author self-citation rate in Italy is 5.65%, while in China, it is 2.52%. We also investigated the distribution of the number of previous papers and journal impact factor across countries (*Figure 3—figure supplement 1*). Self-citation maps by country were highly correlated with maps of the number of previous papers (Spearman's $r$=0.576, p=4.1e-4; 0.654, p=1.8e-5 for First and Last Authors). They were significantly correlated with maps of average impact factor for Last Authors (0.428, p=0.014) but not Last Authors (Spearman's $r$=0.157, p=0.424). Thus, further investigation is necessary with these covariates in a comprehensive model.

## Self-citation rates by subtopic

We next investigated how self-citation rate varies within subfields of Neuroscience research. Based on Scopus abstract data for papers from 2016 to 2020, we developed a topic model using latent Dirichlet allocation (LDA). In LDA, each abstract is modeled as a distribution of topics, and each topic contains probabilities for many different words.

We assigned each paper to the topic with the highest probability to determine 'subtopics' for each paper. The topic number was chosen as 13 with a parameter search (*Figure 4—figure supplement 1*). Based on the most common words of each topic (*Figure 4—figure supplement 2*), we assigned 13 overall themes: (1) Aging and development, (2) Animal models, (3) Cellular, (4) Clinical research, (5) Clinical trials, (6) Dementia, (7) Depression and anxiety, (8) Functional imaging, (9) Mechanistic, (10) Pain, (11) Schizophrenia, (12) Social Neuroscience, (13) Stroke. We then computed self-citation rates for each of these topics (*Figure 4*) as the total number of self-citation in each topic divided by the total number of references in each topic, and results with seven topics are also presented (*Figure 4—figure supplement 3*; *Figure 4—figure supplement 4*).

We generally found that clinical trial research had the highest self-citation rates for First Authors at 6.07% (95% CI: 5.90%, 6.22%), whereas mechanistic research had the lowest self-citation rate at 3.10% (95% CI: 3.05%, 3.15%). For Last Authors, self-citation rates were highest for Dementia research at 10.34% (95% CI: 10.10%, 10.57%) while Social Neuroscience had the lowest self-citation rate at 6.34% (95% CI: 6.25%, 6.42%). For Any Author, Clinical trials once again had the highest self-citation rate at 20.99% (95% CI: 20.59%, 21.28%), and Social Neuroscience had the lowest self-citation rate at 10.71% (95% CI: 10.55%, 10.71%). For Last Author and Any Author self-citation rates, a different number of authors per field may explain the differences in self-citation rates (Spearman's $r$=0.758, p=0.007; $r$=0.736, p=0.009 for Last and Any Authors, respectively). The same relationship did not hold for First Authors (Spearman's $r$=−0.033, p=0.929).

## Self-citation by gender

Several previous works have explored gender differences in self-citation practices. *King et al., 2017* found that men self-cited 70% more than women from 1991 to 2011, but they did not account for

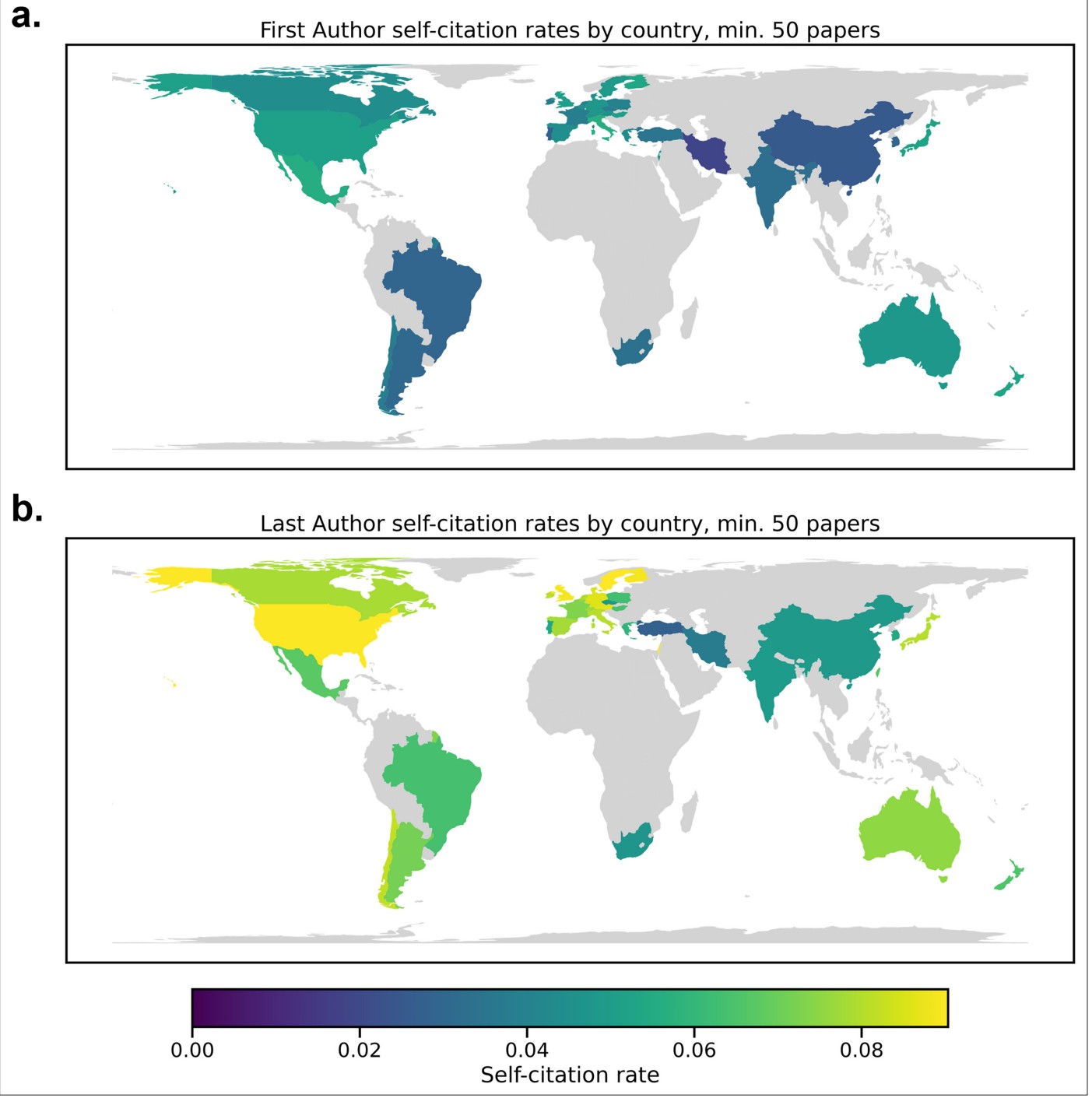

**a.**

First Author self-citation rates by country, min. 50 papers

**b.**

Last Author self-citation rates by country, min. 50 papers

Self-citation rate

**Figure 3.** Self-citation rates by country for First and Last Authors from 2016 to 2020. First Author data are presented in (**a**), and Last Author data are shown in panel (**b**). Only countries with >50 papers were included in the analysis. Country was determined by the affiliation of the author.

The online version of this article includes the following figure supplement(s) for figure 3:

**Figure supplement 1.** Mean impact factor by country for (**a**) First Authors and (**b**) Last Authors.

the number of previous papers that the authors had due to limitations of the dataset. More recent works demonstrated that gender differences in self-citation largely disappear when accounting for the number of possible works an author may self-cite (i.e. number of previous publications) (***Dworkin et al., 2020***; ***Mishra et al., 2018***; ***Azoulay and Lynn, 2020***). While (***Dworkin et al., 2020***) specifically

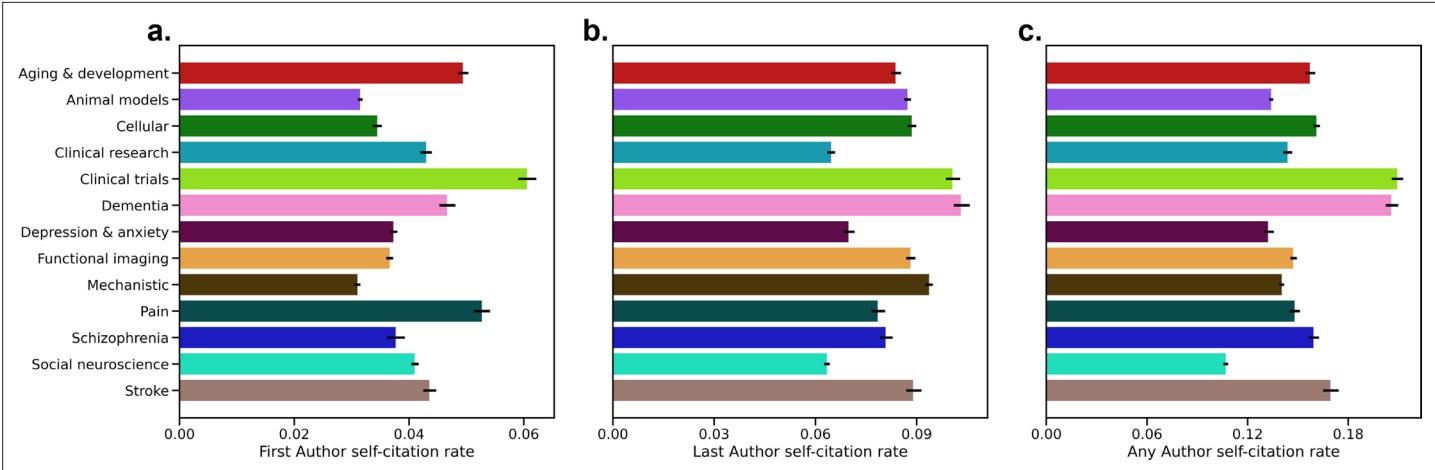

**Figure 4.** Self-citation rates by topic. Results are presented for (**a**) First, (**b**) Last, and (**c**) Any Authors. Topics were determined by Latent Dirichlet Allocation. Confidence intervals of the average self-citation rate are shown based on 1000 iterations of bootstrap resampling.

The online version of this article includes the following figure supplement(s) for figure 4:

**Figure supplement 1.** LDA perplexity on training and validation data for a different number of topics.

**Figure supplement 2.** Topic word clouds for 13 topics.

**Figure supplement 3.** Topic word clouds for seven topics.

**Figure supplement 4.** Self-citation rates by topic for seven topics.

explored citation by gender in the Neuroscience literature, we expand the analysis to a wider range of journals to better represent field-wide self-citation rates (63 journals versus five in the previous work).

For each paper, we assigned a probability of a particular name belonging to a woman or a man using the Genderize.io API. We retained only authors with >80% probabilities. There are clear limitations to these types of packages, as described by *Dworkin et al., 2020*, because they assume genders are binary, and they do not account for authors who identify as nonbinary, transgender, or intersex. As such, the terms 'women' and 'men' indicate the probability of names being that gender as opposed to a specific author identifying as a man or woman. Despite these limitations, we believe these tools can still help broadly uncover gender differences in self-citation rates.

We calculated the proportion of men and women First and Last Authors since 2000 (*Figure 5a*). Although the authorship proportions have begun to converge to be equal by gender, the gender disparity among the Last Authors was more notable than among the First Authors. Men and women were nearly equally represented as First Authors in 2020 (48.60% women). Based on linear fits, we estimated that men and women would be equally represented as Last Authors in 2043 (95% CI: 2040, 2046).

In 2016–2020, there were significant differences between First Author self-citation rates of men and women. First authors who were men had average self-citation rates of 4.54% (95% CI: 3.99%, 5.08%), while women authors had average self-citation rates of 3.39% (95% CI: 3.03%, 3.76%), which is significantly different (p=2.9e-4). Similarly, in 2020, Last Authors who were men had significantly higher self-citation rates than those who were women (p=2.9e-4), with self-citation rates of 8.53% (95% CI: 7.78%, 8.96%) and 7.42% (95% CI: 6.84%, 8.13%), respectively.

In addition, men persistently had higher self-citation rates than women since 2000 (*Figure 5b*), though the gap has slowly decreased. Linear fits were used to estimate that self-citation rates for men and women would be equal for First Authors in the year 2044 (95% CI: 2036, 2056) and equal for the Last Authors in 2040 (95% CI: 2030, 2061). Furthermore, we calculated the ratio of men to women self-citations over the past two decades (*Figure 5c*). For First Authors, men have consistently cited themselves more than women by 27.27–55.57% depending on the year. Among Last Authors, there was a steep decrease in 2002, but since then, men have cited themselves 11.41–43.00% more than women.

Seniority may account for gender differences in self-citation rate, as there are gender disparities in faculty positions and ranks (*Ginther and Hayes, 1999*; *Deutsch and Yao, 2014*; *Li et al., 2021*; *Casad et al., 2021*). To explore the effect of seniority, we investigated self-citation rates by academic age and

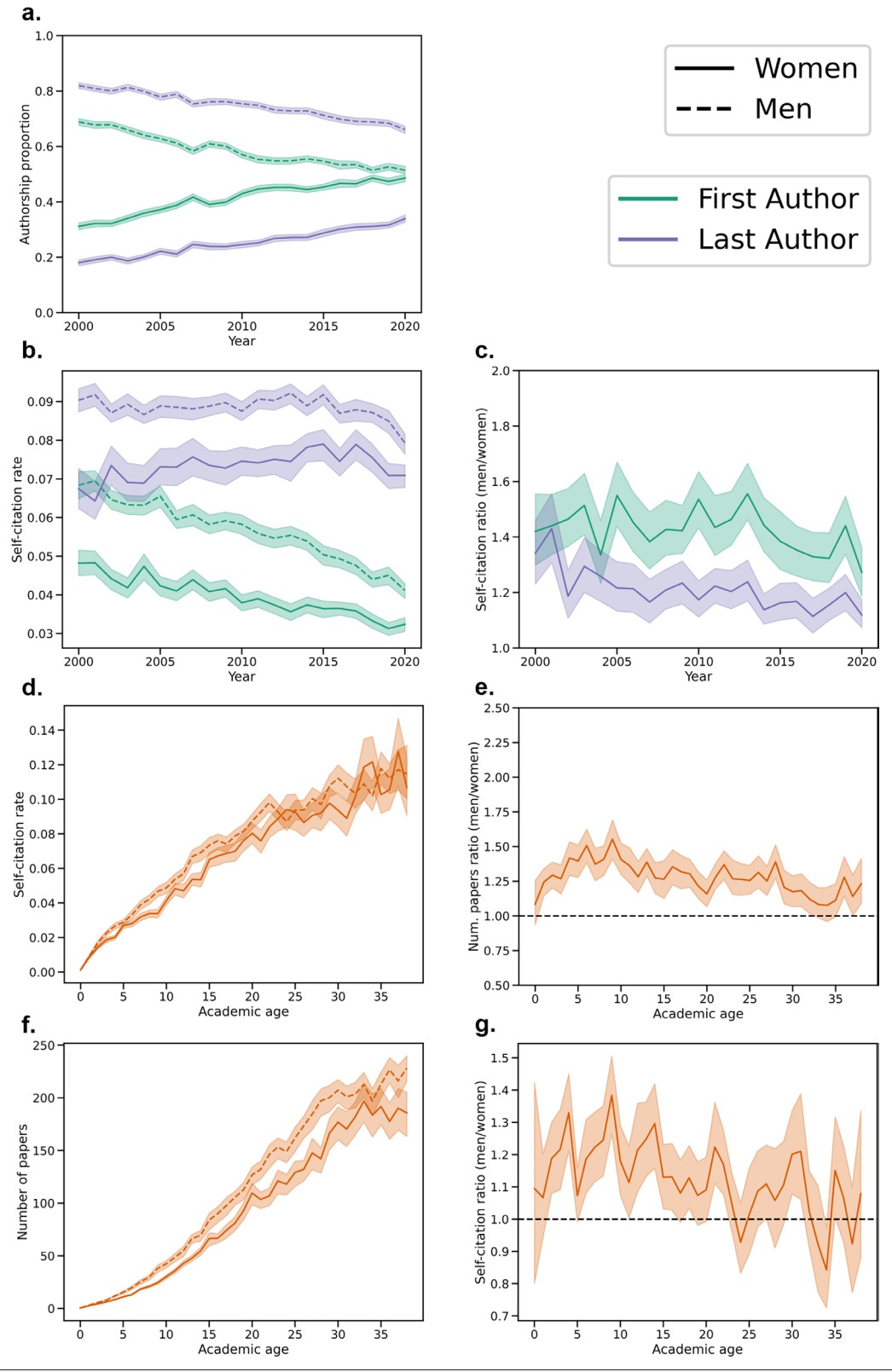

**Figure 5.** Gender disparities in authorship and self-citation. (**a**) Proportion of papers written by men and women First and Last Authors since 2000. (**b**) Average self-citation rates for men and women First and Last Authors. (**c**) Ratio of average self-citation rates of men to women for First and Last Authors. (**d**) Self-citation rates by academic age for men and women authors, where the dashed line represents men and the solid line women. (**e**) Ratio of

*Figure 5 continued on next page*

*Figure 5 continued*

self-citation rates of men to women by academic age. (**f**) Number of papers by academic age for men and women, where the dashed line represents men and the solid line women. (**g**) Ratio of average number of papers of men to women by academic age. In all subplots, 95% confidence intervals of the mean were calculated with 1000 iterations of bootstrap resampling.

The online version of this article includes the following figure supplement(s) for figure 5:

**Figure supplement 1.** Self-citation rates by number of papers for women and men.

**Figure supplement 2.** Topic and gender interactions.

gender (2016–2020). Gender differences for the same academic age emerged early in an academic career and were relatively persistent throughout most of the career (*Figure 5d–e*). For instance, in the previous five years (2016–2020), there were 10,155 papers by early-career women authors and 10,694 by early-career men authors. Women authors had 600,262 references and 13,426 self-citations (2.24% self-citation rate), while men authors had 617,881 references and 18,399 self-citations (2.98% self-citation rate). This equated to a 33.13% higher self-citation rate for men than women during the first ten years of their careers (p=2.9e-4).

We considered two factors that might contribute to the gender discrepancy in self-citation rate by academic age: the number of papers published for authors of a given academic age, which is greater for men at all career stages (*Mishra et al., 2018*; *Azoulay and Lynn, 2020*; *Larivière et al., 2013*; *West et al., 2013*), and the self-citation rate for a given number of papers. We compared the number of papers for men and women at a given academic age (*Figure 5f–g*) and found that men had a higher number of papers. This trend started early in the career (academic age ≤ 10 years), where men had significantly more papers than women (p=2.9e-4). For example, at an academic age of 10 years, men were authors on an average of 42.32 (s.d.: 1.76) papers, and women authored 30.09 (s.d.: 0.96) papers on average. In addition, we divided the number of papers into groups (*Figure 5—figure supplement 1*) and computed self-citation rate by gender for each group. Although the effect was small, men had significantly higher self-citation rates for 0–9 papers (p=7.8e-4) and 10–19 papers (p=0.034). All other differences were not statistically significant. Clearly, accounting for covariates may affect perceived differences in raw self-citation rates. Thus, we further investigate the role of gender by adjusting for various other covariates in Sections *2.9* and *2.10*.

Furthermore, we explored topic-by-gender interactions (*Figure 5—figure supplement 2*). In short, men and women were relatively equally represented as First Authors, but more men were Last Authors across all topics. Self-citation rates were higher for men across all topics.

## Exploring effects of covariates with generalized additive models

Investigating the raw trends and group differences in self-citation rates is important, but several confounding factors may explain some of the differences reported in previous sections. For instance, gender differences in self-citation were previously attributed to men having a greater number of prior papers available to self-cite (*Dworkin et al., 2020*; *Mishra et al., 2018*; *Azoulay and Lynn, 2020*). As such, covarying for various author- and article-level characteristics can improve the interpretability of self-citation rate trends. To allow for inclusion of author-level characteristics, we only consider First Author and Last Author self-citation in these models.

We used generalized additive models (GAMs) to model the number and rate of self-citations for First Authors and Last Authors separately. The data were randomly subsampled so that each author only appeared in one paper. The terms of the model included several article characteristics (article year, average time lag between article and all cited articles, document type, number of references, field, journal impact factor, and number of authors), as well as author characteristics (academic age, number of previous papers, gender, and whether their affiliated institution is in a low- and middle-income country). Model performance (adjusted $R^2$) and coefficients for parametric predictors are shown in *Table 2*. Plots of smooth predictors are presented in *Figure 6*.

First, we considered several career and temporal variables. Consistent with prior works (*Mishra et al., 2018*; *Azoulay and Lynn, 2020*), self-citation rates and counts were higher for authors with a greater number of previous papers. Self-citation counts and rates increased rapidly among the first 25 published papers but then more gradually increased. Early in the career, increasing academic age

**Table 2.** Coefficients and P values for parametric terms in the models.

Separate models were created for First and Last Authors. Models were also made for self-citation counts, self-citation rates, and the number of previously published papers. Quantile-quantile plots are presented in *Figure 6—figure supplement 1*. Results from 100 random resamplings are presented in *Figure 6—figure supplement 2*. Please note that model covariates were not included in the multiple comparisons correction in *Appendix 2—table 5*. *p<0.05, **p<1e-5, ***p<1e-10.

| | | Count | | Rate | | Number of papers | |
|---|---|---|---|---|---|---|---|
| | | First Author | Last Author | First Author | Last Author | First Author | Last Author |
| Adjusted $R^2$ | | 0.508 | 0.351 | 0.347 | 0.204 | 0.565 | 0.400 |
| Deviance explained | | 50.1% | 38.6% | 40.8% | 25.4% | 72.5% | 55.7% |
| Intercept | | 0.046** (p=1.1e-6) | 0.748*** (p<2e-16) | −3.64*** (p<2e-16) | −2.93*** (p<2e-16) | 2.296*** (p<2e-16) | 3.727*** (p<2e-16) |
| | Neurology | −0.093*** (p<2e-16) | −0.025* (p=0.046) | −0.131*** (p<2e-16) | −0.062** (p=1.4e-6) | 0.026* (p=3.7e-4) | 0.068*** (p=4.0e-15) |
| | Neuroscience | 0.147*** (p<2e-16) | 0.184*** (p<2e-16) | 0.112*** (p<2e-16) | 0.186*** (p<2e-16) | −0.195*** (p<2e-16) | −0.122*** (p<2e-16) |
| Field | Psychiatry | 0 | 0 | 0 | 0 | 0 | 0 |
| Low-middle income country status | No | 0 | 0 | 0 | 0 | 0 | 0 |
| | Yes | −0.116** (p=1.1e-7) | −0.241*** (p<2e-16) | −0.127** (p=1.0e-7) | −0.237*** (p<2e-16) | 0.071* (p=2.2e-5) | 0.010 (p=0.605 |
| | Woman | 0 | 0 | 0 | 0 | 0 | 0 |
| Gender | Man | −0.009 (p=0.253) | −0.033* (p=0.002) | −0.026* (p=0.004) | −0.047* (p=5.8e-5) | 0.246*** (p<2e-16) | 0.248*** (p<2e-16) |
| | Article | 0 | 0 | 0 | 0 | 0 | 0 |
| Document type | Review | −0.042**(p=0.001) | −0.139*** (p<2e-16) | −0.064** (p=7.8e-6) | −0.143*** (p<2e-16) | 0.152*** (p<2e-16) | −0.019* (p=0.047) |

was related to greater self-citation. There was a small peak at about five years, followed by a small decrease and a plateau. We found an inverted U-shaped trend for average time lag and self-citations, with self-citations peaking approximately 3 years after initial publication. In addition, self-citations have generally been decreasing since 2000. The smooth predictors showed larger decreases in the First Author model relative to the Last Author model (*Figure 6*).

Then, we considered whether authors were affiliated with an institution in a low- and middle-income country (LMIC). LMIC status was determined by the Organisation for Economic Co-operation and Development. We opted to use LMIC instead of affiliation country or continent to reduce the number of model terms. We found that papers from LMIC institutions had significantly lower self-citation counts (–0.138 for First Authors, –0.184 for Last Authors) and rates (–12.7% for First Authors, –23.7% for Last Authors) compared to non-LMIC institutions. Additional results with affiliation continent are presented in *Appendix 3—table 1*. Relative to the reference level of Asia, higher self-citations were associated with Africa (only three of four models), the Americas, Europe, and Oceania.

Among paper characteristics, a greater number of references was associated with higher self-citation counts and lower self-citation rates (*Figure 6*). Interestingly, self-citations were greater for a small number of authors, although the effect diminished after about five authors. Review articles were associated with lower self-citation counts and rates. No clear trend emerged between self-citations and journal impact factor. In an analysis by field, despite the raw results suggesting that self-citation rates were lower in Neuroscience, GAM-derived self-citations were greater in Neuroscience than in Psychiatry or Neurology. Field-based results were comparable when defining fields by each author's publication history instead of the journal of each article. The most notable difference was in Neuro-science, where authors had relatively higher self-citation rates using author-based rather than journal-based definitions of field (*Appendix 3—table 2*).

Finally, our results aligned with previous findings of nearly equivalent self-citation rates for men and women after including covariates, even showing slightly higher self-citation rates in women. Since raw data showed evidence of a gender difference in self-citation that emerges early in the career but dissipates with seniority, we incorporated two interaction terms: one between gender and academic

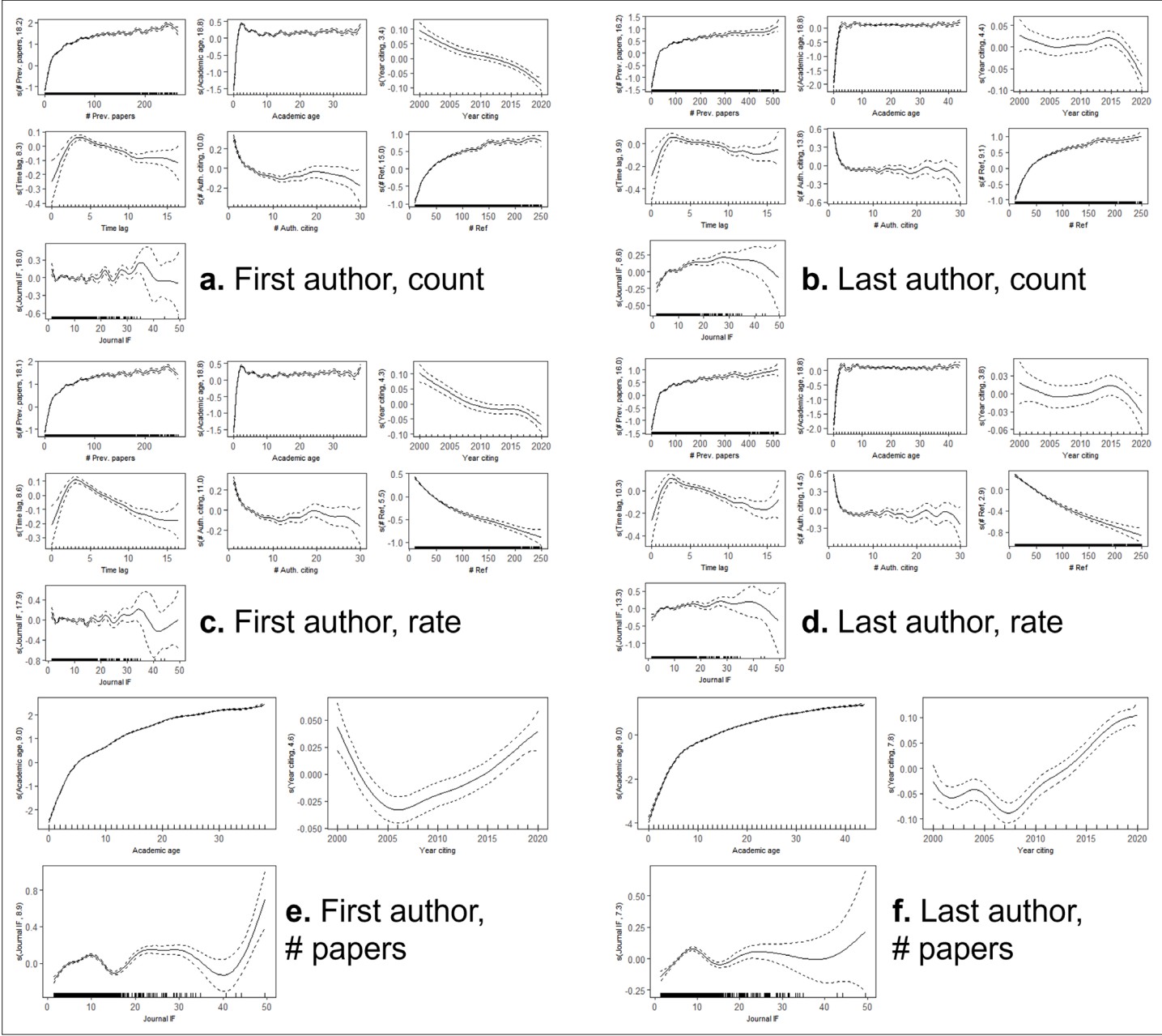

**Figure 6.** Smooth predictors for generalized additive models presented in *Table 2*. Models for (**a**) First Authors and self-citation counts, (**b**) Last Authors and self-citation counts, (**c**) First Authors and self-citation rates, (**d**) Last Authors and self-citation rates, (**e**) First Authors and publication history, (**f**) Last Authors and publication history. The number in parentheses on each y-axis reflects the effective degrees of freedom. All p values were p<2e-16 except year citing for Last Authors for the count (p=5.0e-5) and rate (p=0.176) models.

The online version of this article includes the following figure supplement(s) for figure 6:

**Figure supplement 1.** Quantile-quantile plots for all models.

**Figure supplement 2.** Values for parametric terms in models across 100 random resamplings.

age and a second between gender and the number of previous papers. Results remained largely unchanged with the interaction terms (*Appendix 3—table 3*).

## Reconciling differences between raw data and models

The raw and GAM-derived data exhibited some conflicting results, such as for gender and field of research. To further study covariates associated with this discrepancy, we modeled the publication

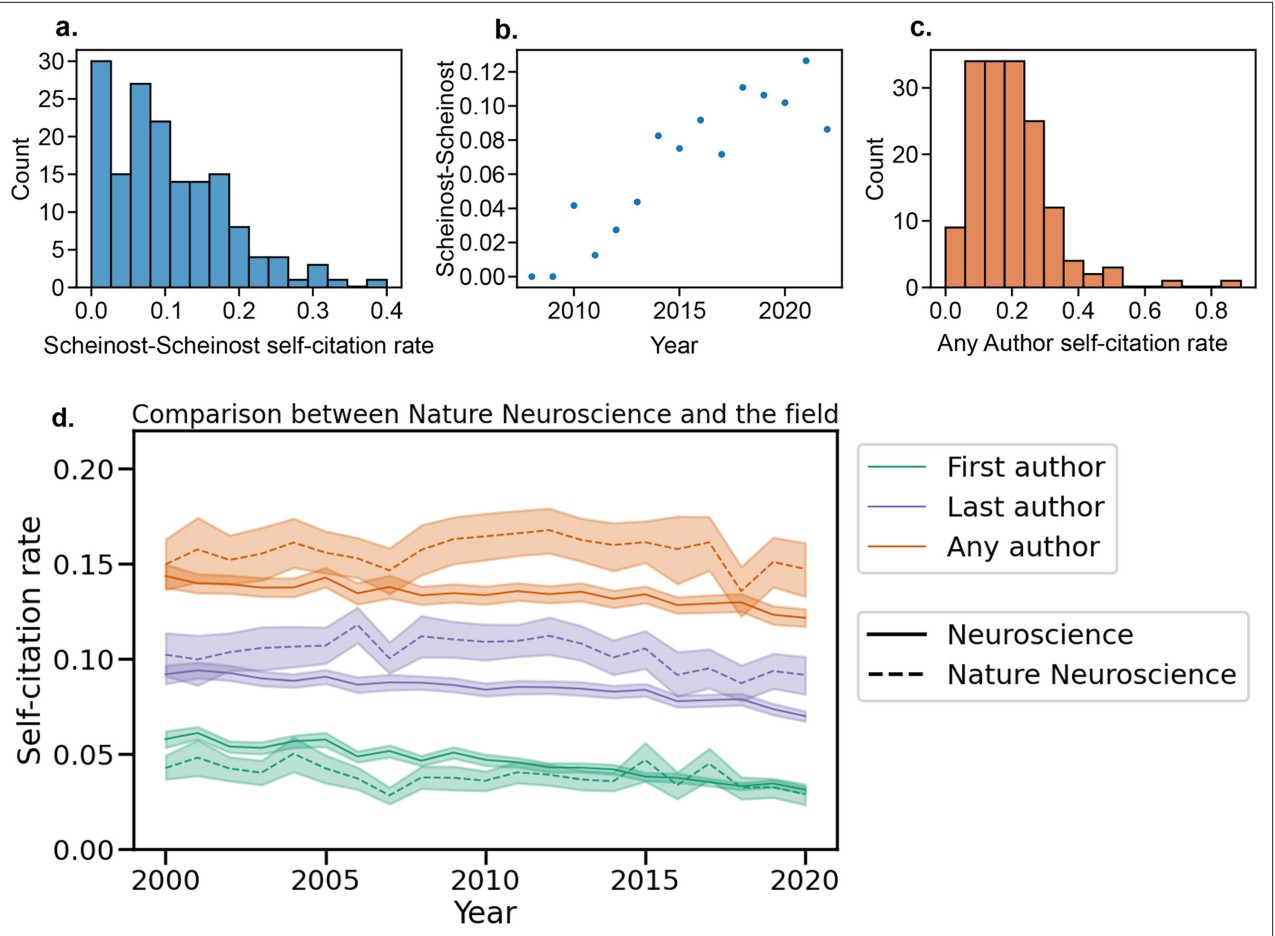

**Figure 7.** Self-citation tool outputs. Self-citation rates for a single author (**a**) across all papers, (**b**) by year, and (**c**) for all authors of their papers. (**d**) Self-citation rate for a single journal is shown compared to the average in Neuroscience from 2000–2020.

history for each author (at the time of publication) in our dataset (*Table 2*). The model terms included academic age, article year, journal impact factor, field, LMIC status, gender, and document type. Notably, Neuroscience was associated with the fewest number of papers per author. This explains how authors in Neuroscience could have the lowest raw self-citation rates by highest self-citation rates after including covariates in a model. In addition, being a man was associated with about 0.25 more papers. Thus, gender differences in self-citation likely emerged from differences in the number of papers, not in any self-citation practices.

## Self-citation code

We provide code for authors to evaluate their own self-citation rates at the following link: https://github.com/mattrosenblatt7/self_citation (copy archived at *Rosenblatt, 2025*). Please note that this code requires access to Scopus, which may be available through your institution. The code may also be adapted for journal editors to evaluate the author self-citation rates of published articles in their journal.

As an example, we also investigated self-citation rates for a particular author, in this case Dustin Scheinost. Dr. Scheinost permitted us to use his name and self-citation data in this work. We show a histogram of self-citations by paper (*Figure 7a*), the self-citation rates over time (*Figure 7b*), and the histogram of Any Author self-citation rates for all of Dr. Scheinost's papers (*Figure 7c*). Furthermore, these methods can be extended to evaluate self-citation rates at the level of a country, institute, or journal. For instance, we compared self-citation rates in *Nature Neuroscience* to the overall field of Neuroscience (*Figure 7d*). In general, Last Author and Any Author self-citation rates were higher in

*Nature Neuroscience* compared to the field. First Author self-citation rates used to be lower in *Nature Neuroscience* (e.g. Year 2000) but are now approximately equal to that of the field.

## Discussion

This work analyzed self-citation rates in 100,347 peer-reviewed Neurology, Neuroscience, and Psychiatry papers, with over five million total citations, to dissect the factors associated with self-citation practices.

### Temporal trends in self-citation rates

Increasing collaborations and expanding author lists in recent years likely explains the increase in Any Author self-citation rates. A more concerning trend is the decrease in First Author relative to Last Author self-citations since 2000. In the Neurosciences, First Authors are typically early-career researchers (e.g. graduate students, postdoctoral fellows) who perform the majority of the experiments and analysis, whereas Last Authors are typically professors who oversee the project and secure funding. As a result, these changes in citation practices could make it harder for early-career scientists to advance in their academic careers, warranting further investigation and monitoring. Another possible explanation is that an increasing number of early career researchers are leaving academia (*Langin, 2022*). Thus, early-career researchers may be less incentivized to self-promote (e.g. self-cite) for academic gains compared to 20 years ago. A third, more optimistic explanation is that principal investigators (typically Last Authors) are increasingly self-citing their lab's papers to build up their trainee's citation records for an increasingly competitive job market.

Differences between early- and late-career researchers' self-citation practices is not surprising because, as one continues in their career, they contribute to more papers and are more likely to cite themselves. In addition, researchers may often become more specialized throughout their career, which may necessitate higher self-citation rates later in the career. However, these results demonstrate a 'snowball effect', whereby senior authors continually accumulate a disproportionate number of self-citations. For example, an author with 30 years of experience cites themselves approximately twice as much as one with 10 years of experience on average. Both authors have plenty of works that they can cite, and likely only a few are necessary. As such, we encourage authors to be cognizant of their citations and to avoid unnecessary self-citations.

### Geographic differences in self-citation rates

There are several possible explanations for differences in self-citation by geographic region, including broader cultural differences or academic culture differences. For instance, an analysis of management journals previously found that self-citation rates of authors from individualist cultures were higher than that of authors from collectivist cultures (*Deschacht and Maes, 2017*). In addition to broader cultural norms affecting the tendency to self-cite, differences in academic norms likely play a major role as well. Researchers in the United States, for example, reported feeling more pressure to publish papers within their organizations compared to researchers from other countries (*van Dalen and Henkens, 2012*). The pressure to publish stems from pressure to advance one's career. Similar pressures that vary by geographic region may drive researchers to unnecessarily self-cite to improve their citation metrics and make them more competitive candidates for hiring, promotion, and funding.

In addition, low- and middle-income countries were associated with fewer self-citations, even after considering numerous covariates. Decreased self-citations may diminish the visibility of researchers from LMIC relative to their peers from non-LMIC. Thus, future research should explore the mechanism behind the decreased self-citations.

While hiring and promotion almost universally depend on citation metrics to some extent, an example of a recent policy in Italy demonstrates how rules regarding hiring and promotion can influence self-citation behavior. This policy was introduced in 2010 and required researchers to achieve certain citation metrics for the possibility of promotion, which was followed by increases of self-citation rates throughout Italy (*Seeber et al., 2019*). Ideally, authors, institutions, journals, and policymakers would work together to establish self-citation guidelines and discourage a 'game the system' mindset. However, requiring all institutions and countries to follow similar values regarding citation metrics is

not practical, so awareness of possible differences in metrics by geographic region due to self-citation differences is the next best alternative.

## Field differences in self-citation rates

Initially, it appeared that self-citation rates in Neuroscience are lower than Neurology and Psychiatry, but after considering several covariates, the self-citation rates are higher in Neuroscience. This discrepancy likely emerges because authors in Neuroscience journals in our dataset tended to be more junior (fewer number of previous papers, slightly lower academic age) compared to Neurology and Psychiatry, giving the illusion of lower field-wide self-citation rates. The field-wide differences in self-citation rate likely depend on both necessity and opportunity. In some research fields, a researcher may need to reference several of their previous works to properly explain the methodology used in the present study, thus having a high necessity of self-citation. Depending on the nature of the work across various fields, researchers may publish more or less frequently, which will affect their number of previous works and thus their opportunity to self-cite.

In addition, while not included in the model to limit the number of terms, the 13 subtopics under examination had different raw self-citation rates, and 'acceptable levels'' of self-citation may vary depending on the subfield. For example, clinical trials had the highest self-citation rate, which may relate to the relatively high number of authors per paper in clinical trial research or the fact that clinical trial research often builds upon previous interventions (e.g. Phase 1 or 2 trials). Overall, these field and subfield differences highlight the importance of editors and researchers understanding common self-citation rates in their specific fields to ensure that they are not unnecessarily self-citing.

## Self-citation rates by gender

The higher self-citation rate of men compared to women, without considering other covariates, aligns with the previous self-citation literature (*Dworkin et al., 2020*; *King et al., 2017*; *Mishra et al., 2018*; *Azoulay and Lynn, 2020*). Similar to prior works (*Dworkin et al., 2020*; *Mishra et al., 2018*; *Azoulay and Lynn, 2020*), we found that the largest difference in self-citing is explained by the number of previous papers (i.e. number of citable items) as opposed to differences in self-citation behavior itself. This result overall points toward a more general underrepresentation of women in science, such as in publication counts (*Larivière et al., 2013*; *West et al., 2013*), collaboration networks (*Zeng et al., 2016*; *Li et al., 2022*), awards (*Melnikoff and Valian, 2019*), editorial boards (*Palser et al., 2022*), and faculty positions (*Smith, 1993*; *Nguyen et al., 2021*; *Wapman et al., 2022*). We confirmed this idea by modeling the number of previous papers for each author. Women had significantly fewer papers than men after considering multiple covariates, such as academic age. In other words, women have a lower self-citation rate than men in the Neuroscience literature because they are not given the same opportunity, such as through prior publications, to self-cite. Establishing field-wide influence and scientific prominence may be most crucial in early career stages, since soon thereafter decisions will be made about hiring, early-career grants, and promotion. Thus, future work should further consider the downstream effects of differences in the number of publications by gender.

## Limitations

There were several notable limitations of this study. First, our analyses were restricted to the top-ranked Neurology, Neuroscience, and Psychiatry journals, and the generalizability of these findings to a wider variety of journals has yet to be determined. Citations of a journal's articles directly affect the journal's impact factor. As such, it is possible that the selection of journals based on high impact factor skews the results toward higher self-citation rates compared to the entire field of Neuroscience. Yet, we found minimal effect of impact factor in our models. Second, we calculated differences between Neurology, Neuroscience, and Psychiatry journals by assigning each journal to only one field (*Appendix 1—table 1*). As some journals publish across multiple fields (e.g. both Neuroscience and Psychiatry research), this categorization provides a gross estimate of differences between fields. Third, we reported averages of self-citation rates across various groups (e.g. academic ages), but there is a wide inter-author and inter-paper variability in self-citation rate. Fourth, as described above, we evaluated gender differences with gender assignment based on name, and this does not account for nonbinary, transgender, or intersex authors. Fifth, selecting subtopics using LDA was subjective because we assigned each subtopic name based on the most common words. Sixth, our modeling techniques are

not useful for prediction due to the inherently large variability in self-citation rates across authors and papers, but they instead provide insight into broader trends. In addition, these models do not account for whether a specific citation is appropriate, as some situations may necessitate higher self-citation rates. Seventh, the analysis presented in this work is not causal. Association studies are advantageous for increasing sample size, but future work could investigate causality in curated datasets. Similarly, this study falls short in several potential mechanistic insights, such as by investigating citation appropriateness via text similarity or international dynamics in authors who move between countries. Yet, this study may lay the groundwork for future works to explore causal estimands (*Lundberg et al., 2021*). Eighth, authors included in this work may not be neurologists, neuroscientists, or psychiatrists. However, they still publish in journals from these fields. Ninth, data were differentially missing (*Appendix 1—table 3*) due to Scopus coverage and gender estimation. Differential missingness could bias certain results in the paper, but we hope that the dataset is large enough to reduce any potential biases. Tenth, while we considered academic age, we did not consider cohort effects. Cohort effects would depend on the year in which the individual started their career. Finally, our analysis does not account for other possible forms of excessive self-citation practices, such as coercive induced self-citation from reviewers (*Ioannidis, 2015*). Despite these limitations, we found significant differences in self-citation rates for various groups, and thus we encourage authors to explore their trends in self-citation rates. Self-citation rates that are higher than average are not necessarily wrong, but suggest that authors should further reflect on their current self-citation practices.

## Self-citation policies

According to The Committee on Publication Ethics (COPE), 'citations where the motivations are merely self promotional…violates publication ethics and is unethical' (*COPE Council, 2019*). Excessive and unnecessary self-citations can possibly be limited by using appropriate citation metrics that cannot be easily 'gamed' (*Seeber et al., 2019*; *Ioannidis, 2015*). Furthermore, while COPE suggests that journals and editors should make policies about acceptable levels of self-citation (*COPE Council, 2019*), many journals have no such policy. For example, only 24.71% of General Surgery *Sanfilippo et al., 2021a* and 14.29% of Critical Care (*Sanfilippo et al., 2021b*) journals had policies regarding self-citation, most of which were policies discouraging 'excessive' or 'inappropriate' self-citations. Although the self-citation policies in the investigated journals had no significant effect on self-citation rate (*Sanfilippo et al., 2021a*; *Sanfilippo et al., 2021b*), a more appropriate consideration might be whether these policies significantly reduce excessive self-citations. Self-citation practices are not typically problematic, but excessive self-citations may falsely establish community-wide influence (*Szomszor et al., 2020*). As such, we believe that the self-citation summary statistics presented in this work could serve as a useful guide in identifying potential cases of excessive self-citation. In practice, there should be more nuance than a binary threshold of acceptable/unacceptable levels of self-citation, as some fields may have atypical self-citation patterns (*Szomszor et al., 2020*) or specific articles may require high levels of self-citation.

## Conclusions

Overall, we identified trends in self-citation rates by time, geographic region, gender, and field, though the extent to which this reflects an underlying problem that needs to be addressed remains an open question. We do not intend to argue against the practice of self-citation, which is not inherently bad and in fact can be beneficial to authors and useful scientifically (*Fowler and Aksnes, 2007*; *Ioannidis, 2015*). Yet, self-citation practices become problematic when they are different across groups or are used to 'game the system'. Future work should investigate the downstream effects of self-citation differences to see whether they impact the career trajectories of certain groups. We hope that this work will help to raise awareness about factors influencing self-citation practices to better inform authors, editors, funding agencies, and institutions in Neurology, Neuroscience, and Psychiatry.

## Methods

We collected data from the 25 journals with the highest impact factors, based on Web of Science impact factors, in each of Neurology, Neuroscience, and Psychiatry. Some journals appeared in the top 25 list of multiple fields (e.g. both Neurology and Neuroscience), so 63 journals were ultimately

included in our analysis. We recognize that limiting the journals to the top 25 in each field also limits the generalizability of the results. However, there are tradeoffs between breadth of journals and depth of information. For example, by limiting the journals to these 63, we were able to look at 21 years of data (2000–2020). In addition, the definition of fields are somewhat arbitrary. By restricting the journals to a set of 63 well-known journals, we ensured that the journals belonged to Neurology, Neuroscience, or Psychiatry research. It is also important to note that the impact factor of these journals has not necessarily always been high. For example, *Acta Neuropathologica* had an impact factor of 17.09 in 2020 but 2.45 in 2000. To further recognize the effects of impact factor, we decided to include an impact factor term in our models.

## Dataset collection

The data were downloaded from the Scopus API in 2021–2022 via http://api.elsevier.com and http://www.scopus.com. We obtained information about research and review articles in the 63 journals from 2000 to 2020. We downloaded two sets of.csv files: (1) an article database and (2) a reference database. For each year/journal, the article database contains last names and first initials of the authors, title, year, and article EID (a unique identifier assigned by Scopus) of all research and review articles. The reference database contains the same information for all articles referenced by any article in the article database.

## Python code using Pybliometrics API

We used the Pybliometrics API (*Rose and Kitchin, 2019*) to access citation information for each entry in the article database. First, we used the article EID to retrieve a detailed author list, which included full names and Scopus Author IDs, and a list of references for each article. For each reference, we extracted the list of Scopus Author IDs. To count as a self-citation, we required that the Scopus Author IDs matched exactly.

Our self-citation metrics included First Author, Last Author, and Any Author self-citation rates. For First (Last) Author self-citation rates, we computed the proportion of reference papers on which the citing First (Last) author is also an author. We considered papers with only a single author as both First Author and Last Author self-citations. For Any Author self-citation rates, we found the proportion of papers for which at least one of the citing authors (any authorship position) was also an author. For the analyses in this paper, we reported total (or weighted average) self-citation rates for different groups. For example, in *Figure 1*, the reported self-citation rate for the year 2000 is the total number of self-citations in 2000 across all papers divided by the total number of references in 2000 across all papers.

Other data we collected from Scopus and Pybliometrics included the affiliation of the authors, the number of papers published by the First and Last Authors before the current paper, and academic age of the First and Last Authors, which we defined as the time between the author's first publication and their current publication.

**Table 3.** Data exclusions.

Each cell shows the number of articles or citations remaining after exclusion, as well as the percentage that were dropped by the exclusion criteria.

| | First Author | | Last Author | |
|---|---|---|---|---|
| | # Articles | # Citations | # Articles | # Citations |
| Prior to exclusions | 157,287 | 8,438,733 | 157,287 | 8,438,733 |
| Missing covariates: remaining (% dropped) | 133,403 (15.18%) | 7,392,638 (12.40%) | 132,806 (15.56%) | 7,379,581 (12.55%) |
| Missing citation data: remaining (% dropped) | 133,256 (0.11%) | 6,773,293 (8.38%) | 132,667 (0.10%) | 6,769,081 (8.27%) |
| Extreme values (citation level): remaining (% dropped) | 126,938 (4.74%) | 6,390,129 (5.66%) | 126,168 (4.90%) | 6,396,015 (5.51%) |
| Extreme values (article level): remaining (% dropped) | 115,205 (9.24%) | 5,794,926 (9.31%) | 114,622 (9.15%) | 5,801,367 (9.30%) |
| Data available for First and Last Authors | 100,347 Articles; 5,061,417 citations | | | |

## Data exclusions and missingness

Data were excluded across several criteria: missing covariates, missing citation data, out-of-range values at the citation pair level, and out-of-range values at the article level (*Table 3*). After downloading the data, our dataset included 157,287 articles and 8,438,733 citations. We excluded any articles with missing covariates (document type, field, year, number of authors, number of references, academic age, number of previous papers, affiliation country, gender, and journal). Of the remaining articles, we dropped any for missing citation data (e.g. cannot identify whether a self-citation is present due to lack of data). Then, we removed citations with unrealistic or extreme values. These included an academic age of less than zero or above 38/44 for First/Last Authors (99th percentile); greater than 266/522 papers for First/Last Authors (99th percentile); and a cited year before 1500 or after 2023. Subsequently, we dropped articles with extreme values that could contribute to poor model stability. These included greater than 30 authors; fewer than 10 references or greater than 250 references; and a time lag of greater than 17 years. These values were selected to ensure that GAMs were stable and not influenced by a small number of extreme values.

In addition, we evaluated whether the data were not missing at random (*Appendix 1—table 3*). Data were more likely to be missing for reviews relative to articles, for Neurology relative to Neuroscience or Psychiatry, in works from Africa relative to the other continents, and for men relative to women. Scopus ID coverage contributed in part to differential missingness. However, our exclusion criteria also contribute. For example, Last Authors with more than 522 papers were excluded to help stabilize our GAMs. More men fit this exclusion criteria than women.

## Country affiliation

For both First and Last Authors, we found the country of their institutional affiliation listed on the publication. In the case of multiple affiliations, the first one listed in Scopus was used. We then calculated the total First Author and Last Author self-citation rate by country, only including countries that had at least 50 First Author or Last Author papers in these select journals from 2016 to 2020. We then projected the self-citation rates onto a map using Geopandas (*Jordahl, 2020*), specifically using the map with coordinate systems EPSG:6933 (https://epsg.io/6933). We determined whether a country was considered a low- and middle-income country based on the Organisation for Economic Co-operation and Development's list (https://wellcome.org/grant-funding/guidance/low-and-middle-income-countries).

## Topic modeling

Latent Dirichlet Allocation (LDA) (*Blei et al., 2003*; *Hoffman et al., 2010*) was implemented with the Gensim package *Rehurek and Sojka, 2010* in Python. LDA is a generative probabilistic model that is commonly used in natural language processing to discover topics in a large set of documents. In LDA, each document is modeled as a distribution of latent topics, and each topic is represented as a distribution of words. Based on the data provided, in this case abstracts from all articles in our dataset from 2016 to 2020, the model finds distributions of topics and words to maximize the log likelihood of the documents. Further details about LDA are available in *Blei et al., 2003*; *Hoffman et al., 2010*; *Rehurek and Sojka, 2010*.

For our implementation, we first removed all special characters and numbers from the abstract data. Then, we lemmatized the words using the Natural Language Toolkit (*Bird et al., 2009*). We excluded words that appeared in less than 20 documents, words that appeared in over 50% of the documents, common stop words (e.g. 'the', 'you', etc.), and some additional words that we felt would not meaningfully contribute to the topic model (e.g. 'associated', 'analysis', 'effect', etc.). In addition, we allowed for bigrams (two consecutive words) and trigrams (three consecutive words) in the model, as long as they appeared at least 20 times in the dataset.

Our total corpus included 41,434 documents with 16,895 unique tokens (words + bigrams + trigrams). We used 90% of the corpus to train our LDA model, and left out 10% to evaluate the perplexity, where a lower perplexity demonstrates better performance, as described in *Blei et al., 2003*. For the a-priori belief on document-topic distribution, we used Gensim's 'auto' option. We trained models with a number of topics ranging from 2 to 20, passing through the entire train corpus 30 times for each number of topics we evaluated. The number of topics was picked based on two evaluation metrics. First, we selected 13 topics as the topics that seemed most meaningful, as assessed

qualitatively by word clouds for each topic. Second, we selected seven topics as the number of topics with the lowest validation perplexity.

Finally, we assigned each paper a discrete topic by choosing the topic with highest probability. Since we do not necessarily care about the generalization of this model and are instead using it to determine topics of a specific set of papers, we determined topics on the same data on which the model was trained.

### Name gender probability estimation

To compute gender probabilities, we submitted given names of all First and Last Authors to the Genderize.io API. Each name was assigned a probability of a name belonging to a woman or man, and we only used names for which Genderize.io assigned at least an 80% probability. Details about the Genderize.io database used to calculate probabilities is available at this link: https://genderize. io/our-data.

There are clear limitations to probabilistically assigning genders to names with packages such as Genderize.io, as described in *Dworkin et al., 2020*, because they assume genders are binary and do not account for authors who identify as nonbinary, transgender, or intersex. As such, the terms 'women' and 'men' indicate the probability of a name being that gender and not that a specific author identifies as a man or woman. However, these tools are still useful to explore broad trends in self-citation rates for women and men.

### Self-citation rate for a particular author

We also calculated the self-citation rate for a particular author, in this case Dr. Dustin Scheinost, in *Figure 5—figure supplement 1*. Here, we defined Scheinost-Scheinost self-citation rates as the proportion of references with Dr. Scheinost as one of the authors. Notably, Dr. Scheinost can be in any author position on the citing or cited article. In *Figure 5—figure supplement 1c*, we calculated the Any Author self-citation rate for all of Dr. Scheinost's papers.

### Confidence intervals

Confidence intervals were computed with 1000 iterations of bootstrap resampling at the article level. For example, of the 100,347 articles in the dataset, we resampled articles with replacement and recomputed all results. The 95% confidence interval was reported as the 2.5 and 97.5 percentiles of the bootstrapped values.

We grouped data into exchangeability blocks to avoid overly narrow confidence intervals or overly optimistic statistical inference. Each exchangeability block comprised any authors who published together as a First Author / Last Author pairing in our dataset. We only considered shared First/Last Author publications because we believe that these authors primarily control self-citations, and otherwise exchangeability blocks would grow too large due to the highly collaborative nature of the field. Furthermore, the exchangeability blocks do not account for co-authorship in other journals or prior to 2000.

### P values

P values were computed with permutation testing using 10,000 permutations, with the exception of regression p values and p values from model coefficients. For comparing different fields (e.g. Neuroscience and Psychiatry) and comparing self-citation rates of men and women, the labels were randomly permuted by exchangeability block to obtain null distributions. For comparing self-citation rates between First and Last Authors, the first and last authorship was swapped in 50% of exchangeability blocks.

In total, we made 40 comparisons (not including the models of self-citation). All p values described in the main text were corrected with the Benjamini/Hochberg (*Benjamini and Hochberg, 1995*) false discovery rate (FDR) correction. With 10,000 permutations, the lowest p value after applying FDR correction is p=2.9e-4, which indicates that the true point would be the most extreme in the simulated null distribution. Further details about each comparison and p values can be found in *Appendix 3— table 4*.

### Exploring effects of covariates with generalized additive models

For these analyses, we used the full dataset size separately for First and Last Authors (*Appendix 2— table 2*). This included 115,205 articles and 5,794,926 citations for First Authors, and 114,622 articles

and 5,801,367 citations for Last Authors. We modeled self-citation counts, self-citation rates, and number of previous papers for First Authors and Last Authors separately, resulting in six total models.

We found that models could be computationally intensive and unstable when including author-level random effects because in many cases there was only one author per group. Instead, to avoid inappropriately narrow confidence bands, we resampled the dataset such that each author was only represented once. For example, if Author A had five papers in this dataset, then one of their five papers was randomly selected. The random resampling was repeated 100 times as a sensitivity analysis (*Figure 6—figure supplement 2*).

For our models, we used generalized additive models from mgcv's 'gam' function in R *Wood, 2017*. The smooth terms included all the continuous variables: number of previous papers, academic age, year, time lag, number of authors, number of references, and journal impact factor. The linear terms included all the categorical variables: field, gender affiliation country LMIC status, and document type. We empirically selected a Tweedie distribution (*Dunn and Smyth, 2005*) with a log link function and $P=1.2$. The p parameter indicates that the variance is proportional to the mean to the p power (*Wood, 2017*). The p parameter ranges from 1 to 2, with $P=1$ equivalent to the Poisson distribution and $P=2$ equivalent to the gamma distribution. For all fitted models, we simulated the residuals with the DHARMa package, as standard residual plots may not be appropriate for GAMs (*Hartig, 2022*). DHARMa scales the residuals between 0 and 1 with a simulation-based approach (*Hartig, 2022*). We also tested for deviation from uniformity, dispersion, outliers, and zero inflation with DHARMa. Non-uniformity, dispersion, outliers, and zero inflation were significant due to the large sample size, but small in effect size in most cases. The simulated quantile-quantile plots from DHARMa suggested that the observed and simulated distributions were generally aligned, with the exception of slight misalignment in the models for the number of previous papers. These analyses are presented in *Figure 6—figure supplement 1* and *Appendix 3—table 4*.

In addition, we tested for inadequate basis functions using mgcv's 'gam.check()' function (*Wood, 2017*). Across all smooth predictors and models, we ultimately selected between 10 and 20 basis functions depending on the variable and outcome measure (counts, rates, papers). We further checked the concurvity of the models and ensured that the worst-case concurvity for all smooth predictors was less than 0.8.

## Journal-based vs. author-based field sensitivity analyses

We refined our field-based analysis to focus only on authors who could be considered neuroscientists, neurologists, and psychiatrists. For each author, we examined the number of articles they had in each subfield, as defined by Scopus. We considered 12 subfields that fell within Neurology, Neuroscience, and Psychiatry, which are presented in *Appendix 1—table 2*. For both First Authors and Last Authors, we excluded them if any of their three most frequently published subfields did not include one of the 12 subfields of interest. If an author's top three subfields included multiple broader fields (e.g., both Neuroscience and Psychiatry), then that author was categorized according to the field in which they published the most articles. Among First Authors, there were 86,220 remaining papers, split between 33,054 (38.33%) in Neurology, 23,216 (26.93%) in Neuroscience, and 29,950 (34.73%) in Psychiatry. Among Last Authors, there were 85,954 remaining papers, split between 31,793 (36.98%) in Neurology, 25,438 (29.59%) in Neuroscience, and 28,723 (33.42%) in Psychiatry.

## Citation diversity statement

Recent work in several fields of science has identified a bias in citation practices such that papers from women and other minority scholars are under-cited relative to the number of such papers in the field (*Bertolero et al., 2020*; *Dworkin et al., 2020*; *Chatterjee and Werner, 2021*; *Fulvio et al., 2021*; *Mitchell et al., 2013*; *Maliniak et al., 2013*; *Caplar et al., 2017*; *Dion et al., 2018*; *Wang et al., 2021*). Here, we sought to proactively consider choosing references that reflect the diversity of the field in thought, form of contribution, gender, race, ethnicity, and other factors. First, we obtained the predicted gender of the First and Last Author of each reference by using databases that store the probability of a first name being carried by a woman (*Dworkin et al., 2020*; *Zhou et al., 2020*). By this measure (and excluding self-citations to the First and Last Authors of our current paper), our references contain 12.53% woman(first)/woman(last), 19.27% man/woman, 13.17% woman/man, and 55.03% man/man. This method is limited in that (a) names, pronouns, and social media profiles used

to construct the databases may not, in every case, be indicative of gender identity and (b) it cannot account for intersex, non-binary, or transgender people. Second, we obtained predicted racial/ethnic category of the First and Last Author of each reference by databases that store the probability of a first and last name being carried by an author of color (*Ambekar et al., 2009*; *Sood and Laohaprapanon, 2018*). By this measure (and excluding self-citations), our references contain 7.46% author of color (first)/author of color(last), 17.45% white author/author of color, 14.81% author of color/white author, and 60.29% white author/white author. This method is limited in that (a) names, Census entries, and Wikipedia profiles used to make the predictions may not be indicative of racial/ethnic identity, and (b) it cannot account for Indigenous and mixed-race authors, or those who may face differential biases due to the ambiguous racialization or ethnicization of their names. We look forward to future work that could help us to better understand how to support equitable practices in science.

## Additional information

### Funding
No external funding was received for this work.

### Author contributions
Matthew Rosenblatt, Conceptualization, Data curation, Software, Formal analysis, Investigation, Visualization, Methodology, Writing – original draft; Saloni Mehta, Hannah Peterson, Data curation, Writing – review and editing; Javid Dadashkarimi, Maya L Foster, Brendan D Adkinson, Qinghao Liang, Violet M Kimble, Jean Ye, Marie C McCusker, Michael C Farruggia, Max J Rolison, Rongtao Jiang, Writing – review and editing; Raimundo Rodriguez, Margaret L Westwater, Conceptualization, Writing – review and editing; Stephanie Noble, Software, Writing – review and editing; Dustin Scheinost, Supervision, Funding acquisition, Writing – review and editing

### Author ORCIDs
Matthew Rosenblatt ⓘ https://orcid.org/0000-0002-3894-6198
Saloni Mehta ⓘ https://orcid.org/0000-0003-1775-4776
Brendan D Adkinson ⓘ https://orcid.org/0000-0003-3196-8674
Marie C McCusker ⓘ https://orcid.org/0000-0003-1067-3375
Michael C Farruggia ⓘ https://orcid.org/0000-0002-5306-6754
Max J Rolison ⓘ https://orcid.org/0000-0001-9534-3767
Dustin Scheinost ⓘ https://orcid.org/0000-0002-6301-1167

Joint Public Review: https://doi.org/10.7554/eLife.88540.4.sa1
Author response https://doi.org/10.7554/eLife.88540.4.sa2

## Additional files

### Supplementary files
MDAR checklist

### Data availability
The data and code are available via GitHub: https://github.com/mattrosenblatt7/self_citation (copy archhived at *Rosenblatt, 2025*). The data were downloaded from the Scopus API in 2021–2022 via http://api.elsevier.com and http://www.scopus.com. The shared dataset has been anonymized such that specific articles cannot be identified. In addition, the GitHub repository includes code to gather self-citation data about yourself, with appropriate access to Scopus.

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

# Appendix 1

## Study information

**Appendix 1—table 1.** All journals included in this analysis by field, sorted alphabetically. We categorized each journal as belonging to Neurology, Neuroscience, or Psychiatry. While we recognize that some journals belong to overlapping fields (e.g., Neurology and Neuroscience), we attempted to select the most relevant field for each journal.

| Field | Journals (2020 Impact Factor) |
|---|---|
| Neurology | Acta Neuropathologica (17.09); Alzheimer's and Dementia (21.57); Alzheimer's Research and Therapy (6.98); Annals of Neurology (10.42); Brain (13.50); Brain Stimulation (8.96); Epilepsy Currents (7.5); JAMA Neurology (18.30); JNNP (10.28); Journal of Headache and Pain (7.28); Journal of Stroke (6.97); Lancet Neurology (44.18); Molecular Neurodegeneration (14.20); Movement Disorders (10.34); Nature Reviews Neurology (42.94); Neuro-Oncology (12.30); Neurology (9.91); Neurology: Neuroimmunology and NeuroInflammation (8.49); Neuropathology and Applied Neurobiology (8.09); Neurotherapeutics (7.62); npj Parkinson's Disease (8.65); Pain (6.96); Sleep Medicine Reviews (11.61); Stroke (7.91); Translational Stroke Research (6.83) |
| Neuroscience | Annual Review of Neuroscience (12.45); Behavioral and Brain Sciences (12.58); Brain, Behavior, and Immunity (7.22); Frontiers in Neuroendocrinology (8.61); Journal of Neuroinflammation (8.32); Journal of Pineal Research (13.01); Nature Human Behaviour (13.66); Nature Neuroscience (24.88); Nature Reviews Neuroscience (34.87); Neuron (17.17); Neuroscience and Biobehavioral Reviews (8.99); Neuroscientist (7.52); Progress in Neurobiology (11.69); Trends in Cognitive Sciences (20.23); Trends in Neurosciences (13.84) |
| Psychiatry | Acta Psychiatrica Scandinavica (6.39); Addiction (6.53); American Journal of Psychiatry (18.11); Biological Psychiatry (13.38); Bipolar Disorders (6.74); Body Image (6.41); British Journal of Psychiatry (9.32); Clinical Psychological Science (7.17); Depression and Anxiety (6.51); Epidemiology and Psychiatric Sciences (6.89); Evidence-Based Mental Health (8.54); JAACAP (8.83) JAMA Psychiatry (21.60); JCPP (8.98); Journal of Abnormal Psychology (6.67); Journal of Behavioral Addictions (6.76); Molecular Psychiatry (15.99); Neuropsychopharmacology (7.86); Psychological Medicine (7.72); Psychotherapy and Psychosomatics (17.66); Schizophrenia Bulletin (9.31); The Lancet Psychiatry (26.48); World Psychiatry (49.55) |

**Appendix 1—table 2.** Mapping of subfields to fields.

| Field | Scopus-defined Subfields |
|---|---|
| Neurology | Neurology; Neurology (clinical) |
| Neuroscience | Cognitive Neuroscience; Neuroscience (all); Cellular and Molecular Neuroscience; Behavioral Neuroscience; Neuropsychology and Physiological Psychology; Developmental Neuroscience; Neuroscience (miscellaneous) |
| Psychiatry | Biological Psychiatry; Psychiatric Mental Health; Psychiatry and Mental Health |

**Appendix 1—table 3.** Data missingness.

| | | Ratio of prevalence in missing to non-missing data | |
|---|---|---|---|
| | | First Author | Last Author |
| Document type | Article | 0.994 | |
| | Review | 1.029 | |
| Field | Neurology | 1.204 | |
| | Neuroscience | 0.888 | |
| | Psychiatry | 0.900 | |

*Appendix 1—table 3 Continued on next page*

*Appendix 1—table 3 Continued*

|  |  | Ratio of prevalence in missing to non-missing data | |
| --- | --- | --- | --- |
|  | Africa | 1.308 | 1.329 |
|  | Americas | 0.973 | 0.979 |
|  | Asia | 1.562 | 1.570 |
|  | Europe | 0.909 | 0.908 |
| Continent | Oceania | 0.926 | 0.914 |
|  | No | 0.972 | 0.976 |
| Low-middle income country status | Yes | 1.615 | 1.608 |
|  | Woman | 0.864 | 0.922 |
| Gender | Man | 1.089 | 1.026 |

# Appendix 2

## Additional descriptive analyses

**Appendix 2—table 1.** Comparisons of self-citation rates whether defining field by paper or by author.

| Field | Field definition | First Author | Last Author |
|---|---|---|---|
| | By journal | 4.54 (4.36, 4.70) | 8.87 (8.52, 9.14) |
| Neurology | By author | 4.33 (4.14, 4.47) | 9.07 (8.71, 9.36) |
| | By journal | 3.41 (3.30, 3.51) | 7.54 (7.36, 7.73) |
| Neuroscience | By author | 3.62 (3.47, 3.74) | 8.32 (8.13, 8.51) |
| | By journal | 4.29 (4.11, 4.43) | 8.41 (8.16, 8.60) |
| Psychiatry | By author | 4.45 (4.24, 4.60) | 7.92 (7.58, 8.16) |

**Appendix 2—table 2.** Percentiles of self-citation rates in articles from 2016 to 2020.

| Percentile | First Author self-citation rate (%) | Last Author self-citation rate (%) | Any Author self-citation rate (%) |
|---|---|---|---|
| 1% | 0.00 | 0.00 | 0.00 |
| 5% | 0.00 | 0.00 | 0.00 |
| 10% | 0.00 | 0.00 | 2.38 |
| 25% | 0.00 | 2.44 | 6.67 |
| 50% | 2.86 | 7.14 | 13.51 |
| 75% | 7.69 | 13.79 | 22.72 |
| 90% | 15.00 | 21.95 | 33.33 |
| 95% | 20.83 | 28.21 | 41.18 |
| 99% | 35.71 | 41.94 | 58.33 |

**Appendix 2—table 3.** Correlations between year and self-citation rate and corresponding slopes by field.

| | | Correlation | Slope (% per decade) |
|---|---|---|---|
| | First Author | −0.86 (-0.92,−0.77) | −0.71 (-0.87,−0.54) |
| | Last Author | 0.43 (0.09, 0.67) | 0.30 (0.05, 0.53) |
| Neurology | Any Author | 0.87 (0.80, 0.93) | 1.68 (1.19, 2.08) |
| | First Author | −0.96 (-0.98,−0.94) | −1.40 (-1.51,−1.28) |
| | Last Author | −0.90 (-0.95,−0.85) | −0.94 (-1.10,−0.77) |
| Neuroscience | Any Author | −0.82 (-0.91,−0.70) | −0.80 (-1.06,−0.56) |
| | First Author | −0.95 (-0.97,−0.92) | −1.30 (-1.48,−1.15) |
| | Last Author | 0.51 (0.28, 0.68) | 0.36 (0.17, 0.53) |
| Psychiatry | Any Author | 0.66 (0.41, 0.80) | 0.76 (0.36, 1.06) |

**Appendix 2—table 4.** First Author and Last Author self-citation rates by affiliation country of the author for papers from 2016–2020.

95% confidence intervals obtained via bootstrap resampling are included in parentheses. Only countries with at least 50 papers were included in the analysis.

| Country | First Author Self-citation Rate | Last Author Self-citation Rate |
| --- | --- | --- |
| Argentina | 3.04 (2.59, 3.42) | 7.11 (5.72, 8.35) |
| Australia | 4.82 (4.51, 5.07) | 7.54 (6.96, 7.93) |
| Austria | 4.62 (3.68, 5.20) | 8.73 (7.24, 9.62) |
| Belgium | 4.61 (4.10, 5.04) | 7.58 (6.58, 8.21) |
| Brazil | 2.92 (2.60, 3.21) | 6.37 (5.54, 6.98) |
| Canada | 4.43 (4.23, 4.61) | 7.85 (7.55, 8.13) |
| Chile | 3.79 (2.87, 4.67) | 8.37 (5.37, 9.57) |
| China | 2.52 (2.31, 2.74) | 4.84 (4.51, 5.20) |
| Czech Republic | 3.84 (2.64, 4.93) | 4.85 (3.67, 6.16) |
| Denmark | 4.45 (4.07, 4.76) | 8.51 (7.69, 9.09) |
| Finland | 5.34 (4.82, 5.79) | 8.86 (8.08, 9.56) |
| France | 3.83 (3.63, 4.01) | 7.32 (6.97, 7.62) |
| Germany | 4.79 (4.63, 4.95) | 8.61 (8.37, 8.83) |
| Greece | 4.36 (3.63, 5.05) | 5.91 (4.56, 6.99) |
| Hong Kong | 4.72 (3.32, 5.87) | 6.83 (5.74, 8.15) |
| Hungary | 5.10 (4.03, 5.98) | 6.44 (5.31, 7.55) |
| India | 3.29 (2.50, 3.96) | 5.00 (3.77, 5.89) |
| Ireland | 3.67 (3.20, 4.11) | 8.12 (6.93, 8.96) |
| Iran | 1.87 (1.24, 2.42) | 3.78 (2.40, 4.90) |
| Israel | 4.68 (4.20, 5.11) | 9.00 (8.16, 9.70) |
| Italy | 5.65 (5.35, 5.90) | 8.08 (7.57, 8.46) |
| Japan | 5.25 (4.87, 5.55) | 8.05 (7.59, 8.43) |
| South Korea | 2.93 (2.50, 3.28) | 5.47 (4.92, 5.95) |
| Mexico | 5.92 (3.56, 7.21) | 7.01 (4.76, 8.11) |
| Netherlands | 3.97 (3.81, 4.16) | 7.92 (7.41, 8.29) |
| New Zealand | 5.34 (4.44, 6.11) | 6.52 (5.60, 7.31) |
| Norway | 4.90 (4.23, 5.39) | 8.83 (7.43, 9.88) |
| Poland | 3.98 (3.27, 4.63) | 6.31 (5.21, 7.36) |
| Portugal | 2.85 (2.31, 3.26) | 5.42 (4.39, 6.27) |
| Singapore | 3.80 (2.60, 4.77) | 7.54 (4.23, 9.13) |
| South Africa | 3.44 (2.47, 4.40) | 4.77 (3.79, 5.89) |
| Spain | 4.47 (4.20, 4.72) | 7.83 (7.35, 8.25) |
| Sweden | 4.89 (4.53, 5.24) | 9.03 (8.66, 9.42) |
| Switzerland | 4.55 (4.26, 4.85) | 7.72 (7.31, 8.18) |
| Taiwan | 4.17 (3.07, 5.01) | 6.66 (4.62, 8.02) |
| Turkey | 3.51 (2.72, 4.18) | 2.79 (2.20, 3.38) |

*Appendix 2—table 4 Continued on next page*

*Appendix 2—table 4 Continued*

| Country | First Author Self-citation Rate | Last Author Self-citation Rate |
|---|---|---|
| United Kingdom | 5.02 (4.84, 5.18) | 8.88 (8.57, 9.10) |
| United States | 5.09 (4.99, 5.17) | 8.97 (8.84, 9.08) |

**Appendix 2—table 5.** *P* values for all 44 comparisons performed in this study.

*P* values are corrected for multiple comparisons with the Benjamini/Hochberg false discovery rate (FDR) correction with $\alpha$=0.05. For *P* values determined by permutation testing, 10,000 permutations were used. Significant values ($P_{corrected}$ <0.05) are marked with an asterisk in the "Finding" column.

| Comparison | Method | Uncorrected *P* | Corrected *P* | Finding |
|---|---|---|---|---|
| First vs Last Author self-citation (all fields) | permutation | 1e-4 | 2.9e-4 | * Last >First |
| First vs Last Author self-citation (Neurology) | permutation | 1e-4 | 2.9e-4 | * Last >First |
| First vs Last Author self-citation (Neuroscience) | permutation | 1e-4 | 2.9e-4 | * Last >First |
| First vs Last Author self-citation (Psychiatry) | permutation | 1e-4 | 2.9e-4 | * Last >First |
| First Author: Neurology vs. Neuroscience | permutation | 1e-4 | 2.9e-4 | * Neurology >Neuroscience |
| First Author: Neuroscience vs. Psychiatry | permutation | 1e-4 | 2.9e-4 | * Psychiatry >Neuroscience |
| First Author: Neurology vs. Psychiatry | permutation | 0.095 | 0.144 | No significant difference |
| Last Author: Neurology vs. Neuroscience | permutation | 1e-4 | 2.9e-4 | * Neurology >Neuroscience |
| Last Author: Neuroscience vs. Psychiatry | permutation | 1e-4 | 2.9e-4 | * Psychiatry >Neuroscience |
| Last Author: Neurology vs. Psychiatry | permutation | 0.078 | 0.123 | No significant difference |
| Any Author: Neurology vs. Neuroscience | permutation | 1e-4 | 2.9e-4 | * Neurology >Neuroscience |
| Any Author: Neuroscience vs. Psychiatry | permutation | 1e-4 | 2.9e-4 | * Psychiatry >Neuroscience |
| Any Author: Neurology vs. Psychiatry | permutation | 0.005 | 0.010 | * Neurology >Psychiatry |
| Slope over the years: First Author | correlation | 2.1e-15 | 9.2e-14 | * $m$=−1.21 % / decade |
| Slope over the years: Last Author | correlation | 0.074 | 0.123 | No significant correlation |
| Slope over the years: Any Author | correlation | 0.012 | 0.024 | No significant correlation |
| Country-level self-citation rate and number of previous papers: First Author | correlation | 1.5e-4 | 4.1e-4 | *Spearman's *r*=0.576 |
| Country-level self-citation rate and number of previous papers: Last Author | correlation | 8.0e-7 | 1.8e-5 | *Spearman's *r*=0.654 |
| Country-level self-citation rate and impact factor: First Author | correlation | 0.347 | 0.424 | No significant correlation |
| Country-level self-citation rate and impact factor: Last Author | correlation | 0.007 | 0.014 | *Spearman's *r*=0.428 |
| First Author: Spearman's correlation between topic self-citation and number of authors | correlation | 0.915 | 0.929 | No significant correlation |
| Last Author: Spearman's correlation between topic self-citation and number of authors | correlation | 0.003 | 0.007 | *Spearman's *r*=0.758 |
| Any Author: Spearman's correlation between topic self-citation and number of authors | correlation | 0.004 | 0.009 | *Spearman's *r*=0.736 |
| Men vs. Women, First Author self-citation rate, 2020 | permutation | 1e-4 | 2.9e-4 | * Men >Women |
| Men vs. Women, Last Author self-citation rate, 2020 | permutation | 4e-4 | 0.001 | * Men >Women |
| Early career men vs. women, self-citation rate | permutation | 1e-4 | 2.9e-4 | * Men >Women |
| Early career men vs. women, number of papers | permutation | 1e-4 | 2.9e-4 | * Men >Women |

*Appendix 2—table 5 Continued on next page*

*Appendix 2—table 5 Continued*

| Comparison | | Method | Uncorrected *P* | Corrected *P* | Finding |
|---|---|---|---|---|---|
| Early career men vs. women self-citation rate by number of papers | 0–9 papers | permutation | 3.0e-4 | 7.8e-4 | * Men >Women |
| | 10–19 | permutation | 0.019 | 0.034 | * Men >Women |
| | 20–29 | permutation | 0.174 | 0.248 | No significant difference |
| | 30–39 | permutation | 0.855 | 0.918 | No significant difference |
| | 40–49 | permutation | 0.035 | 0.062 | No significant difference |
| | 50–59 | permutation | 0.888 | 0.929 | No significant difference |
| | 60–69 | permutation | 0.508 | 0.588 | No significant difference |
| | 70–79 | permutation | 0.272 | 0.342 | No significant difference |
| | 80–89 | permutation | 0.175 | 0.248 | No significant difference |
| | 90–99 | permutation | 0.399 | 0.475 | No significant difference |
| | 100–149 | permutation | 0.929 | 0.929 | No significant difference |
| | 150–199 | permutation | 0.824 | 0.906 | No significant difference |
| | 200–249 | permutation | 0.264 | 0.342 | No significant difference |
| | 250–299 | permutation | 0.196 | 0.269 | No significant difference |
| | 300–399 | permutation | 0.264 | 0.342 | No significant difference |
| | 400–499 | permutation | 0.716 | 0.808 | No significant difference |
| | ≥ 500 | permutation | 0.075 | 0.123 | No significant difference |

# Appendix 3

## Additional model-based analyses

**Appendix 3—table 1.** Models with affiliation continent instead of low- and middle-income country terms.

*P<0.05, **P<1e-5, ***P<1e-10.

| | | Count | | Rate | |
|---|---|---|---|---|---|
| | | First Author | Last Author | First Author | Last Author |
| Adjusted $R^2$ | | 0.507 | 0.354 | 0.347 | 0.208 |
| Deviance explained | | 50.1% | 38.9% | 40.9% | 25.7% |
| Intercept | | −0.096** (P=4.3e-8) | 0.467*** (P<2e-16) | −3.817*** (P<2e-16) | −3.222*** (P<2e-16) |
| Field | Neurology | −0.089*** (P<2e-16) | −0.021 (P=0.098) | −0.124*** (P<2e-16) | −0.058** (P=6.4e-6) |
| | Neuroscience | 0.150*** (P<2e-16) | 0.200*** (P<2e-16) | 0.120*** (P<2e-16) | 0.204*** (P<2e-16) |
| | Psychiatry | 0 | 0 | 0 | 0 |
| Continent | Africa | 0.162 (P=0.069) | 0.211* (P=0.027) | 0.290* (P=0.001) | 0.357* (P=2.1e-4) |
| | Americas | 0.125*** (P=3.1e-15) | 0.309*** (P<2e-16) | 0.162*** (P<2e-16) | 0.320*** (P<2e-16) |
| | Asia | 0 | 0 | 0 | 0 |
| | Europe | 0.162*** (P<2e-16) | 0.256*** (P<2e-16) | 0.198*** (P<2e-16) | 0.270*** (P<2e-16) |
| | Oceania | 0.170*** (P=4.7e-12) | 0.187** (P=1.7e-10) | 0.231*** (P<2e-16) | −.234*** (P=5.0e-14) |
| Gender | Woman | 0 | 0 | 0 | 0 |
| | Man | −0.003 (P=0.703) | −0.024* (P=0.026) | −0.017 (P=0.059) | −0.036* (P=0.002) |
| Document type | Article | 0 | 0 | 0 | 0 |
| | Review | −0.047** (P=1e-4) | −0.139*** (P<2e-16) | −0.073** (P=9.7e-7) | −0.146*** (P<2e-16) |

**Appendix 3—table 2.** Coefficients for field when defining fields based on the publication history of authors rather than the journal.

*P<0.05, **P<1e-5, ***P<1e-10.

| Field | Count | | Rate | | Number of papers | |
|---|---|---|---|---|---|---|
| | First Author | Last Author | First Author | Last Author | First Author | Last Author |
| Neurology (by journal) | −0.093*** (P<2e-16) | −0.025* (P=0.046) | −0.131*** (P<2e-16) | −0.062** (P=1.4e-6) | 0.026* (P=3.7e-4) | 0.068*** (P=4.0e-15) |
| Neurology (by author) | −0.091*** (P=2.9e-16) | −0.002 (P=0.85) | −0.154*** (P<2e-16) | −0.054* (P=2.2e-4) | −0.016* (P=0.034) | 0.042* (P=1.7e-5) |
| Neuroscience (by journal) | 0.147*** (P<2e-16) | 0.184*** (P<2e-16) | 0.112*** (P<2e-16) | 0.186*** (P<2e-16) | −0.195*** (P<2e-16) | −0.122*** (P<2e-16) |
| Neuroscience (by author) | 0.248*** (P<2e-16) | 0.357*** (P<2e-16) | 0.191*** (P<2e-16) | 0.312*** (P<2e-16) | −0.340*** (P<2e-16) | −0.253*** (P<2e-16) |
| Psychiatry (by journal) | 0 | 0 | 0 | 0 | 0 | 0 |
| Psychiatry (by author) | 0 | 0 | 0 | 0 | 0 | 0 |

**Appendix 3—table 3.** Models with interaction terms for between gender/academic age and gender/number of previous papers.
*P<0.05, **P<1e-5, ***P<1e-10.

| | | Count | | Rate | | Number of papers | |
| --- | --- | --- | --- | --- | --- | --- | --- |
| | | First Author | Last Author | First Author | Last Author | First Author | Last Author |
| Adjusted R² | | 0.509 | 0.353 | 0.349 | 0.204 | 0.565 | 0.4 |
| Deviance explained | | 50.1% | 38.6% | 40.9% | 25.4% | 72.5% | 55.7% |
| Intercept | | 0.034* (P=0.001) | 0.748*** (P<2e-16) | –3.645*** (P<2e-16) | –2.926*** (P<2e-16) | 2.306*** (P<2e-16) | 3.724*** (P<2e-16) |
| Field | Neurology | –0.094*** (P<2e-16) | –0.026* (P=0.045) | –0.132*** (P<2e-16) | –0.062** (P=1.3e-6) | 0.026* (P=3.8e-4) | 0.068*** (P=3.8e-15) |
| | Neuroscience | 0.146*** (P<2e-16) | 0.185*** (P<2e-16) | 0.112*** (P<2e-16) | 0.186*** (P<2e-16) | –0.195*** (P<2e-16) | –0.122*** (P<2e-16) |
| | Psychiatry | 0 | 0 | 0 | 0 | 0 | 0 |
| Low-middle income country status | No | 0 | 0 | 0 | 0 | 0 | 0 |
| | Yes | –0.118** (P=7.4e-8) | –0.242*** (P<2e-16) | –0.128** (P=8.0e-8) | –0.237*** (P<2e-16) | 0.073* (P=1.4e-5) | 0.009 (P=0.628) |
| Gender | Woman | 0 | 0 | 0 | 0 | 0 | 0 |
| | Man | 0.019 (P=0.107) | –0.031* (P=0.023) | –0.001 (P=0.911) | –0.048* (P=0.001) | 0.223*** (P<2e-16) | 0.254*** (P<2e-16) |
| Document type | Article | 0 | 0 | 0 | 0 | 0 | 0 |
| | Review | –0.040* (P=0.001) | –0.139*** (P<2e-16) | –0.063* (P=1.3e-5) | –0.142*** (P<2e-16) | 0.151*** (P<2e-16) | –0.019* (P=0.046) |

**Appendix 3—table 4.** Tests for uniformity, outliers, and dispersion in models.

Tests were performed using the DHARMa package in R. Uniformity: Asymptotic one-sample Kolmogorov-Smirnov test. DHARMa outlier test based on exact binomial test with approximate expectations. DHARMa nonparametric dispersion test via sd of residuals fitted vs. simulated. DHARMa zero-inflation test via comparison to expected zeros with simulation under H0=fitted model.

| | Count | | Rate | | Number of papers | |
| --- | --- | --- | --- | --- | --- | --- |
| | First Author | Last Author | First Author | Last Author | First Author | Last Author |
| Uniformity | D=0.010 (P=3.1e-6) | D=0.016 (P=1.4e-9) | D=0.030 (P<2.2e-16) | D=0.041 (P<2.2e-16) | D=0.097 (P<2.2e-16) | D=0.078 (P<2.2e-16) |
| Outliers | 0.009 outlier frequency (P=1.2e-5) | 0.010 outlier frequency (P=5.0e-4) | 0.011 outlier frequency (P=4.0e-14) | 0.009 outlier frequency (P=0.004) | 0.013 outlier frequency (P<2.2e-16) | 0.012 outlier frequency (P<2.2e-16) |
| Dispersion | dispersion = 1.358 (P<2.2e-16) | dispersion = 1.211 (P<2.2e-16) | dispersion = 1.251 (P<2.2e-16) | dispersion = 1.058 (P<2.2e-16) | dispersion = 1.775 (P<2.2e-16) | dispersion = 1.258 (P<2.2e-16) |
| Zero Inflation | ratio = 0.977 (P<2.2e-16) | ratio = 0.858 (P<2.2e-16) | ratio = 0.913 (P<2.2e-16) | ratio = 0.806 (P<2.2e-16) | ratio = 0.250 (P<2.2e-16) | ratio = 0.173 (P<2.2e-16) |

