## [Editor Report · eLife Assessment]

This study examines how self-citations in selected neurology, neuroscience, and psychiatry journals differ according to seniority, geography, gender and subfield. The evidence supporting the claims is **convincing**, and the article is a **valuable** addition to the literature on self-citations.

---

## [Referee Report · Joint Public Review]

Editors’ note: This is the third version of this article, and it addresses the points made during the peer review of the second version by performing additional analyses and clarifying some of the limitations of the study.

Comments made during the peer review of the first version, along with author's responses to these comments, are available with previous versions of the article.

The following summary of the article is taken from comments made by Reviewer #1 about version 2 of the article:

In this manuscript, the authors use a large dataset of neuroscience publications to elucidate the nature of self-citation within the neuroscience literature. The authors initially present descriptive measures of self-citation across time and author characteristics; they then produce an inclusive model to tease apart the potential role of various article and author features in shaping self-citation behavior. This is a valuable area of study, and the authors approach it with a rich dataset and solid methodology.

---

## [Author Response]

The following is the authors’ response to the previous reviews

**Public Reviews:**

**Reviewer#1 (Public review):**
In this manuscript, the authors use a large dataset of neuroscience publications to elucidate the nature of self-citation within the neuroscience literature. The authors initially present descriptive measures of self-citation across time and author characteristics; they then produce an inclusive model to tease apart the potential role of various article and author features in shaping self-citation behavior. This is a valuable area of study, and the authors approach it with a rich dataset and solid methodology.The revisions made by the authors in this version have greatly improved the validity and clarity of the statistical techniques, and as a result the paper's findings are more convincing.This paper's primary strengths are: (1) its comprehensive dataset that allows for a snapshot of the dynamics of several related fields; (2) its thorough exploration of how self-citation behavior relates to characteristics of research and researchers.

Thank you for your positive view of our paper and for your previous comments.

Its primary weakness is that the study stops short of digging into potential mechanisms in areas where it is potentially feasible to do so - for example, studying international dynamics by identifying and studying researchers who move between countries, or quantifying more or less 'appropriate' self-citations via measures of abstract text similarity.

We agree that these are limitations of the existing study. We updated the limitations section as follows (page 15, line 539):

“Similarly, this study falls short in several potential mechanistic insights, such as by investigating citation appropriateness via text similarity or international dynamics in authors who move between countries.”

Yet while these types of questions were not determined to be in scope for this paper, the study is quite effective at laying the important groundwork for further study of mechanisms and motivations, and will be a highly valuable resource for both scientists within the field and those studying it.
**Reviewer#2 (Public review):**
The study presents valuable findings on self-citation rates in the field of Neuroscience, shedding light on potential strategic manipulation of citation metrics by first authors, regional variations in citation practices across continents, gender differences in early-career self-citation rates, and the influence of research specialization on self-citation rates in different subfields of Neuroscience. While some of the evidence supporting the claims of the authors is solid, some of the analysis seems incomplete and would benefit from more rigorous approaches.

Thank you for your comments. We have addressed your suggestions presented in the “Recommendations for the authors” section by performing your recommended sensitivity analysis that specifically identifies authors who could be considered neurologists, neuroscientists, and psychiatrists (as opposed to just papers that are published in these fields). Please see the “Recommendations for the authors” section for more details.

**Reviewer#3 (Public review):**
This paper analyses self-citation rates in the field of Neuroscience, comprising in this case, Neurology, Neuroscience and Psychiatry. Based on data from Scopus, the authors identify self-citations, that is, whether references from a paper by some authors cite work that is written by one of the same authors. They separately analyse this in terms of first-author self-citations and last-author self-citations. The analysis is well-executed and the analysis and results are written down clearly. The interpretation of some of the results might prove more challenging. That is, it is not always clear what is being estimated.This issue of interpretability was already raised in my review of the previous revision, where I argued that the authors should take a more explicit causal framework. The authors have now revised some of the language in this revision, in order to downplay causal language. Although this is perfectly fine, this misses the broader point, namely that it is not clear what is being estimated. Perhaps it is best to refer to Lundberg et al. (2021) and ask the authors to clarify "What is your Estimand?" In my view, the theoretical estimands the authors are interested in are causal in nature. Perhaps the authors would argue that their estimands are descriptive. In either case, it would be good if the authors could clarify that theoretical estimand.

Thank you for your comment and for highlighting this insightful paper. After reading this paper, we believe that our theoretical estimand is descriptive in nature. For example, in the abstract of our paper, we state: “This work characterizes self-citation rates in basic, translational, and clinical Neuroscience literature by collating 100,347 articles from 63 journals between the years 2000-2020.” This goal seems consistent with the idea of a descriptive estimand, as we are not interested in any particular intervention or counterfactual at this stage. Instead, we seek to provide a broad characterization of subgroup differences in self-citations such that future work can ask more focused questions with causal estimands.

Our analysis included subgroup means and generalized additive models, both of which were described as empirical estimands for a theoretical descriptive estimand in Lundberg et al. We added the following text to the paper (page 3, line 112):

“Throughout this work, we characterized self-citation rates with descriptive, not causal, analyses. Our analyses included several theoretical estimands that are descriptive 17, such as the mean self-citation rates among published articles as a function of field, year, seniority, country, and gender. We adopted two forms of empirical estimands. First, we showed subgroup means in self-citation rates. We then developed smooth curves with generalized additive models (GAMs) to describe trends in self-citation rates across several variables.”

In addition, we added to the limitations section as follows (page 15, line 539):

“Yet, this study may lay the groundwork for future works to explore causal estimands.”

Finally, in my previous review, I raised the issue of when self-citations become "problematic". The authors have addressed this issue satisfactorily, I believe, and now formulate their conclusions more carefully.

Thank you for your previous comments. We agree that they improved the paper.

Lundberg, I., Johnson, R., & Stewart, B. M. (2021). What Is Your Estimand? Defining the Target Quantity Connects Statistical Evidence to Theory. American Sociological Review, 86(3), 532-565. https://doi.org/10.1177/00031224211004187

**Recommendations for the authors:**

**Reviewer#1 (Recommendations for the authors):**
Thank you for your thorough revisions and responses to the reviews
**Reviewer#2 (Recommendations for the authors):**
I appreciate the authors' responses and am satisfied with all their replies except for my second comment. I still find the message conveyed slightly misleading, as the results seem to be generalized to neurologists, neuroscientists, and psychiatrists. It is important to refine the analysis to focus specifically on neuroscientists, identified as first or last authors based on their publication history. This approach is common in the science of science literature and would provide a more accurate representation of the findings specific to neuroscientists, avoiding the conflation with other related fields. This refinement could serve as a robustness check in the supplementary. I think adding this sub-analysis is essential to the validity of the results claimed in this paper.

Thank you for your comment. We added a sensitivity analysis where fields are defined by an author’s publication history, not by the journal of each article.

In the main text, we added the following:

(Page 3, line 129) “When determining fields by each author’s publication history instead of the journal of each article, we observed similar rates of self-citation (Table S7). The 95% confidence intervals for each field definition overlapped in most cases, except for Last Author self-citation rates in Neuroscience (7.54% defined by journal vs. 8.32% defined by author) and Psychiatry (8.41% defined by journal vs. 7.92% defined by author).”

Further details are provided in the methods section (page 21, line 801):

“4.11 Journal-based vs. author-based field sensitivity analyses

We refined our field-based analysis to focus only on authors who could be considered neuroscientists, neurologists, and psychiatrists. For each author, we looked at the number of articles they had in each subfield, as defined by Scopus. We considered 12 subfields that fell within Neurology, Neuroscience, and Psychiatry. These subfields are presented in Table S12. For each First Author and Last Author, we excluded them if any of their three most frequently published subfields did not include one of the 12 subfields of interest. If an author’s top three subfields included multiple broader fields (e.g., both Neuroscience and Psychiatry), then that author was categorized according to the field in which they published the most articles. Among First Authors, there were 86,220 remaining papers, split between 33,054 (38.33%) in Neurology, 23,216 (26.93%) in Neuroscience, and 29,950 (34.73%) in Psychiatry. Among Last Authors, there were 85,954 remaining papers, split between 31,793 (36.98%) in Neurology, 25,438 (29.59%) in Neuroscience, and 28,723 (33.42%) in Psychiatry.”

**Reviewer#3 (Recommendations for the authors):**
I would like to thank the authors for their responses the points that I raised, I do not have any new comments or further responses.